# Enhanced homeostatic sleep response and decreased neurodegenerative proteins in cereblon knock-out mice
Jun-Hyung Jung[1,4], Jinhong Kim[2,4], Uroos Akber[1], Na Young Lee[1], Jeong-won Baek[1], Jieun Jung[2], Mincheol Park[2], Jiseung Kang[2], Seungje Jeon[3], Chul-Seung Park [1] ✉ & Tae Kim [2] ✉

Energy homeostasis and sleep have a bidirectional relationship. Cereblon (CRBN) regulates energy levels by ubiquitinating the AMP-activated protein kinase(AMPK), an energy sensor. However, whether CRBN participates in sleep is unclear. Here, we examine sleep–wake patterns in $Crbn^{+/+}$ and $Crbn^{-/-}$ mice during 24-h baseline, 6-h sleep deprivation (SD), and following 6-h recovery sleep (RS). At baseline, overall sleep patterns are similar between genotypes. However, SD decreases CRBN expression in $Crbn^{+/+}$ mice and increases phospho-Tau, phospho-α-synuclein, DNAJA1 (DJ2), and DNAJB1 (DJ1) in both genotypes, with $Crbn^{-/-}$ mice showing a lesser extent of increase in p-Tau and p-α-synuclein and a higher level of heat shock protein 70 (HSP70), DJ2, and DJ1. During RS, $Crbn^{-/-}$ mice show increased slow-wave activity in the low-delta range (0.5–2.5 Hz), suggesting higher homeostatic sleep propensity associated with AMPK hyperactivation. By illuminating the role of CRBN in regulating sleep–wake behaviors through AMPK, we suggest CRBN as a potential therapeutic target for managing sleep disorders and preventing neurodegeneration.

Sleep is a universal phenomenon in living animals[1]. However, it is still unclear why this phenomenon has been conserved throughout evolution. Recent studies have shown the role of sleep in recharging energy stores within the brain. It has been suggested that sleep restores energy balance by ATP production, and wakefulness demands energy for energy-related mechanisms, including unfolded protein response and electron transport chain[2–4]. In particular, the body can conserve energy through a reduction in body temperature during sleep or an increase in daily sleep duration[5].

The two major sleep states—rapid eye movement (REM) and non-REM (NREM)—are as distinct from one another as each is from wakefulness. The electroencephalography (EEG) pattern in NREM sleep is commonly described as synchronous, involving waveforms such as sleep spindles, K-complexes, and high-voltage slow waves. By contrast, REM sleep is defined by EEG activation, muscle atonia and twitches, and episodic REM bursts often accompanied by cardiorespiratory irregularities. In addition, in humans, REM sleep is associated with dreaming based on the recall of vivid dreams after ~80% of arousals from this sleep state[6].

Adenosine triphosphate (ATP) is an intracellular energy source in the brain and other tissues. Adenosine monophosphate (AMP)-activated protein kinase (AMPK) is a potent energy sensor activated when the AMP/ATP ratio increases. Interestingly, AMPK activity and sleep–wake regulation reciprocally affect each other, as demonstrated by an increase in ATP levels during sleep leading to the deactivation of AMPK, and conversely, ATP depletion, which can elevate AMPK activity, increasing the duration of both NREM sleep and slow-wave activity. ATP levels in the brain increase by ~2.5-fold during sleep, leading to deactivation of AMPK[3]. Conversely, depletion of ATP by blocking ATP synthesis, which can elevate AMPK activity, increases the duration of both NREM sleep and slow-wave activity during NREM sleep[7]. In addition, pharmacological manipulation of AMPK bidirectionally affects the slow-wave activity during NREM sleep in mice[8].

Cereblon (CRBN), a direct target of immune-modulatory imide drugs (IMiDs) such as thalidomide, consists of a CRL4$^{CRBN}$ E3 ubiquitin ligase complex with cullin-4A/B, DNA-damage binding protein-1 (DDB1), and RING-box protein-1[9]. CRBN is a substrate receptor of the CRL4$^{CRBN}$ complex, which mediates ubiquitination and successive degradation of its endogenous substrates, such as glutamine synthetase in Alzheimer's disease[10] and AMPK in high-fat diet-induced obesity through the ubiquitin-proteasome system[11–14]. Additionally, structural modification of CRBN through the binding of IMiDs promotes its recruitment and ubiquitination of new substrates, including Sal-like protein 4 in teratogenicity[15] and oncogenic casein kinase 1A1 in myelodysplastic syndrome[16], leading to their degradation.

[1]School of Life Sciences, Gwangju Institute Science and Technology (GIST), Gwangju, Republic of Korea. [2]Department of Biomedical Science and Engineering, GIST, Gwangju, Republic of Korea. [3]Department of Ophthalmology, University of Texas Southwestern Medical Center, Dallas, TX, USA. [4]These authors contributed equally: Jun-Hyung Jung, Jinhong Kim. ✉e-mail: cspark@gist.ac.kr; tae-kim@gist.ac.kr

CRBN is ubiquitously expressed in most rodent organs, including the brain, especially the cerebral cortex, cerebellum, thalamus, and hippocampal region[17]. Since CRBN was first reported as a target gene responsible for a mild form of autosomal recessive non-syndromic mental retardation[18,19], the role of CRBN in the brain has been widely studied. CRBN knock-out ($Crbn^{-/-}$) mice show upregulated the large-conductance, voltage, and calcium-activated potassium (BK) channels activity and reduced excitatory synaptic transmission, resulting in impaired synaptic and cognitive function[20,21]. In addition, CRBN deficiency disrupts synaptic plasticity in mice by altering the AMPK/ mammalian target of the rapamycin signaling cascade, which is responsible for postsynaptic protein synthesis[22,23]. Whereas, hyperactivation of AMPK reduces synaptic function in $Crbn^{-/-}$ mice, it also protects against hypoxia-induced focal cerebral ischemic injury through activation of the protein kinase R (PKR)-like endoplasmic reticulum kinase/eukaryotic translation initiation factor 2A (eIF2α) pathway[24]. Moreover, our group previously found that lack of CRBN relieves tauopathies, including conformation of cytotoxic tau-aggregation in Alzheimer's disease, by enhancing the chaperone activity of DNAJA1 (DJ2)/heat shock protein 70 (Hsp70)[25].

Given that CRBN is critical for the ubiquitination of AMPK, it might be involved in sleep regulation and brain energetics. However, there is a paucity of research on the link between CRBN and sleep. Therefore, we

hypothesized that CRBN regulates sleep–wake behaviors by controlling ATP and AMPK. We sought to investigate the role of CRBN in sleep physiology using CRBN knockout mice.

## Results

### Effect of sleep deprivation (SD) on CRBN expression in $Crbn^{+/+}$ mice

We found that 6 h of SD decreased the amount of CRBN protein in $Crbn^{+/+}$ mice, as evidenced by western blot analysis (Fig. 1a, b). By contrast, amounts of phosphorylated Tau (p-Tau) and phosphorylated α-synuclein (p-α-Syn), which are aggregated forms of those proteins, increased after SD (Fig. 1c–e). Tau and p-Tau bands in western blot could be shown as multiple bands caused by post-translational phosphorylation and aggregation (Figs. 1, 2, and 4 and Supplementary Fig. 3). Furthermore, amounts of the chaperone HSP70 and two co-chaperones, DJ2 and DJ1, increased after SD (Fig. 1f). Consistently, immunohistochemistry analysis showed that SD reduced CRBN and increased p-Tau, phosphorylated α-synuclein, DJ2, and DJ1 in the hippocampal region and thalamus (Fig. 1i, Supplementary Fig. 1). In addition, whereas SD decreased CRBN, a DDB1-CUL4A-associated WD40 domain protein (DCAF), it did not affect two other DCAFs—DCAF1 and DCAF2 (see Supplementary Fig. 2), suggesting that the effect of SD on the expression of substrate receptors for CRL4 is specific to CRBN.

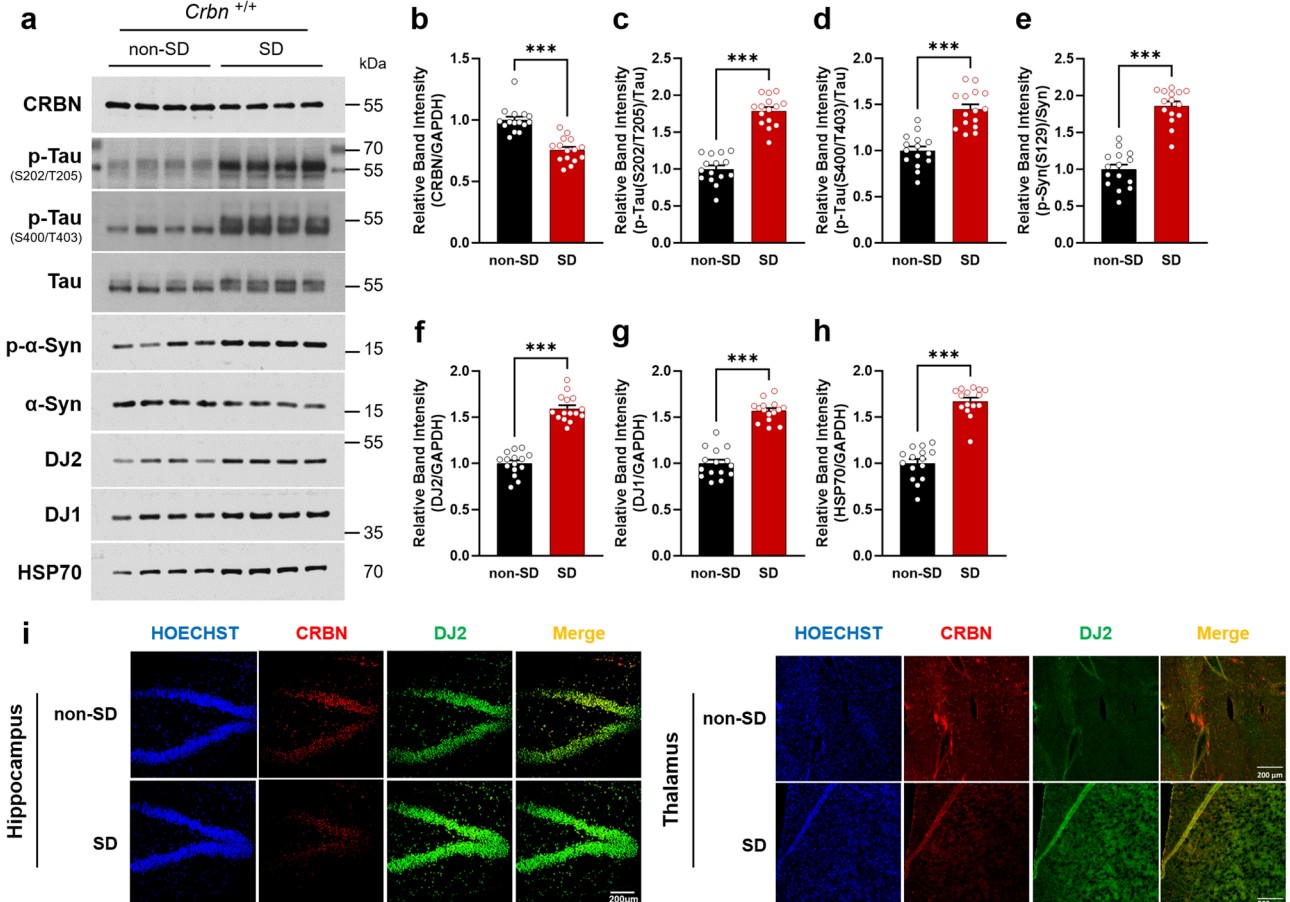

**Fig. 1 | Effects of SD on CRBN expression in $Crbn^{+/+}$ mice. a** Protein levels of CRBN, neuropathological proteins (Tau and α-synuclein(α-Syn)), and a molecular chaperone (HSP70) and co-chaperones (DJ2 and DJ1) in brain lysates ($n = 15$ in each non-SD group and SD group) were measured by western blot analysis after 6-h of SD. **b–h** Band intensity relative to GAPDH, Tau, or α-synuclein was calculated using densitometry. **i** Immunohistochemistry analysis of the hippocampal and

thalamus region was performed using confocal microscopy. Tissues were stained with Hoechst (blue), CRBN (red), and DJ2 (green) in (**i**), and tissue samples stained with p-Tau, DJ2, p-α-synuclein, and DJ1 were illustrated in Supplementary Fig. 1. Representative images from independent experiments are depicted and data are presented as mean ± standard error of the mean (SEM) by unpaired $t$-test. The $P$ values were presented as $^{*}P < 0.05$, $^{**}P < 0.01$, $^{***}P < 0.001$.

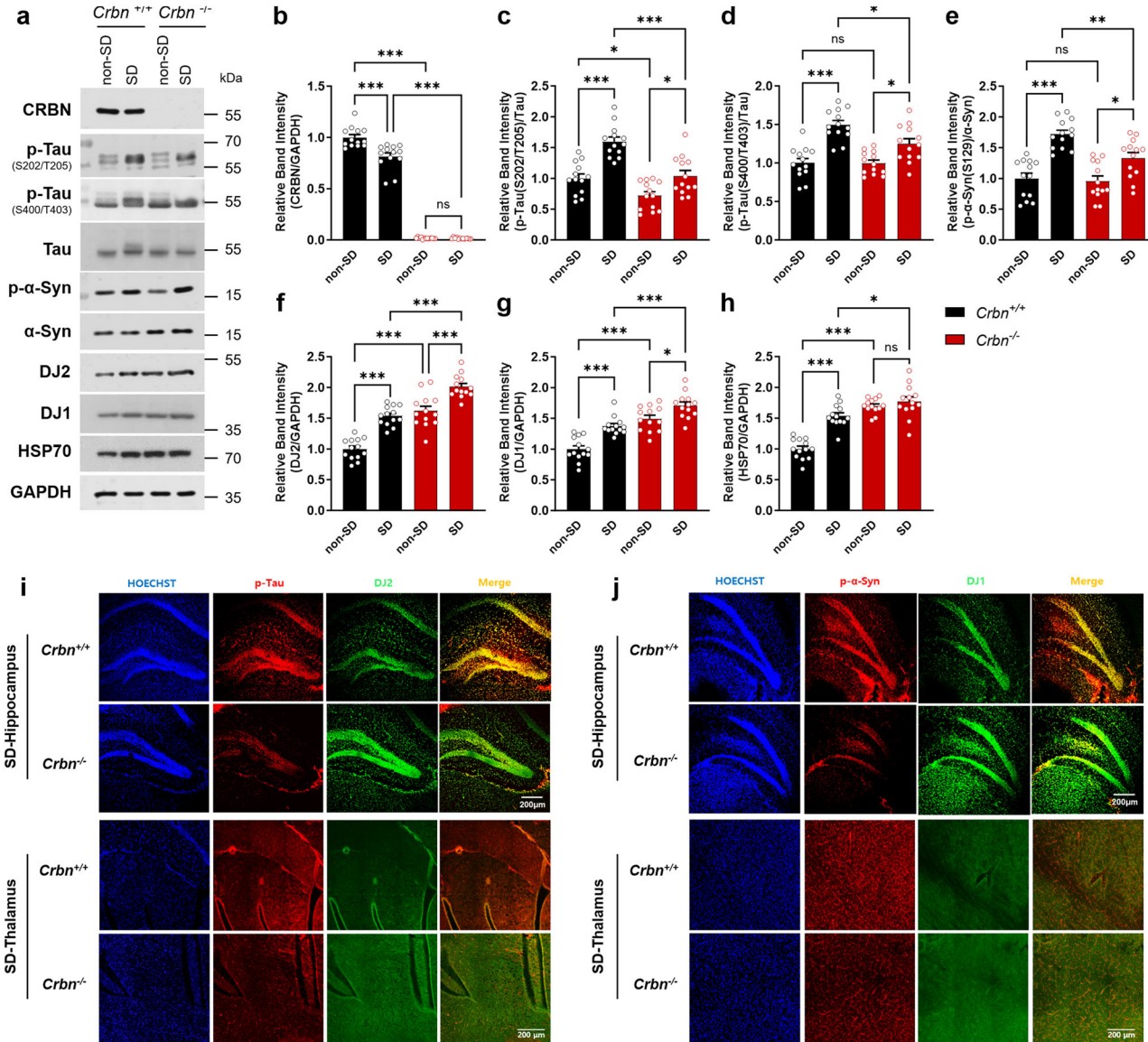

**Fig. 2 | Altered protein expression of stress markers in sleep-deprived *Crbn*<sup>−/−</sup> mice. a** Neuropathological protein, and chaperone expression after 6-h of SD were measured by western blot analysis (*n* = 15 in each *Crbn*<sup>+/+</sup> non-SD, *Crbn*<sup>+/+</sup> SD, *Crbn*<sup>−/−</sup> non-SD, *Crbn*<sup>−/−</sup> SD group). Supplementary Fig. 3 shows the continuous change of those neuropathological proteins and chaperone expression in (**a**) over 6-h SD. **b–h** Band intensity relative to GAPDH, Tau, or α-synuclein was calculated using densitometry. **i–j** Immunohistochemistry analysis of p-Tau (red) and DJ2 (green) or phosphorylated α-synuclein (p-α-Syn, red) and DJ1 (green) in hippocampus and thalamus region. Representative images from independent experiments are depicted, and data are presented as mean ± SEM by two-way repeated measure ANOVA with the Tukey post hoc method. The *P* values were presented as $^*P < 0.05$, $^{**}P < 0.01$, $^{***}P < 0.001$, or ns no significant difference.

## Altered levels of stress markers in *Crbn*<sup>−/−</sup> mice after SD

To further examine the role of CRBN in sleep regulation, we collected brains from sleep-deprived *Crbn*<sup>−/−</sup> mice and performed western blot analysis (Fig. 2a). In the non-SD state, protein level of p-Tau (pS202/T205) was lower in *Crbn*<sup>−/−</sup> mice than in *Crbn*<sup>+/+</sup> mice, however, levels of DJ2, DJ1, and HSP70 were significantly higher in *Crbn*<sup>−/−</sup> mice than in *Crbn*<sup>+/+</sup> mice (Fig. 2b–h). Moreover, *Crbn*<sup>−/−</sup> mice had a lower SD-induced increase in p-Tau and phosphorylated α-synuclein than *Crbn*<sup>+/+</sup> mice. Total amounts of Tau and α-synuclein did not change significantly after SD in *Crbn*<sup>−/−</sup> mice. Also, SD further increased the levels of DJ2 and DJ1 in *Crbn*<sup>−/−</sup> mice, which were already at higher levels in the non-SD state than in *Crbn*<sup>+/+</sup> mice. Also, the same patterns of neurodegenerative proteins and chaperone proteins in the brains of both genotypes during hourly SD were confirmed in Supplementary Fig. 3. Consistently, immunohistochemistry analysis showed that sleep-deprived *Crbn*<sup>−/−</sup> mice had lower levels of p-Tau and phosphorylated α-synuclein and higher levels of DJ2 and DJ1 in each brain region than sleep-deprived *Crbn*<sup>+/+</sup> mice (Fig. 2i, j). Taken together, these results suggest that *Crbn*<sup>−/−</sup> mice are less vulnerable to the effects of SD than *Crbn*<sup>+/+</sup> mice.

## Twenty-four-hour sleep–wake profiles of *Crbn*<sup>+/+</sup> and *Crbn*<sup>−/−</sup> mice

The hourly profiles of wakefulness, NREM sleep, and REM sleep during the spontaneous sleep–wake cycle did not differ between *Crbn*<sup>−/−</sup> and *Crbn*<sup>+/+</sup> mice (Fig. 3a). The percentage of time spent in wakefulness, NREM sleep, and REM sleep during the 24-h or light and dark periods did not show any differences (Fig. 3b). Bout counts for each vigilance state showed no difference between genotypes (Fig. 3c). REM bout length was longer, especially during the light period (Fig. 3d). In summary, despite some differences in sleep parameters between genotypes, *Crbn*<sup>−/−</sup> mice did not show significant alterations in sleep–wake profile across a 24-h period compared to *Crbn*<sup>+/+</sup> mice except REM bout length.

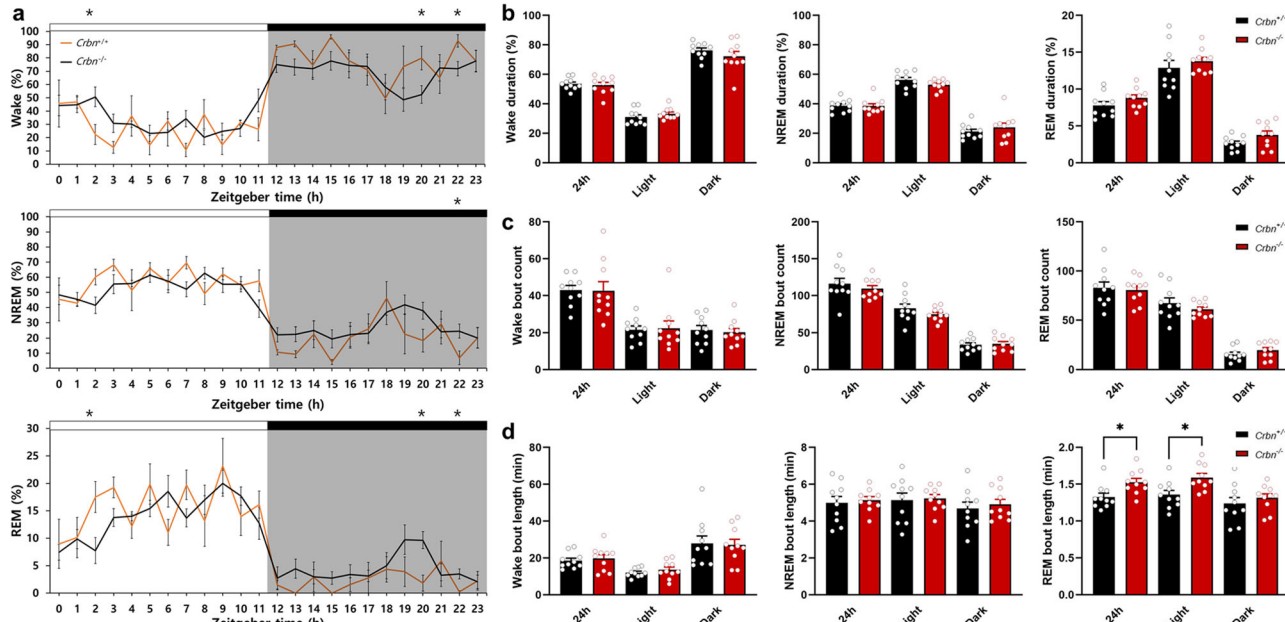

**Fig. 3 | Twenty-four hour sleep–wake profiles of *Crbn*[+/+] and *Crbn*[−/−] mice.**
**a** Proportions of time in wakefulness, NREM sleep, and REM sleep as assessed by EEG recordings during 12-h light and dark periods. Black and red lines represent the mean value at each time point for *Crbn*[+/+] (*n* = 10) and *Crbn*[−/−] (*n* = 10) mice, respectively. Data are presented as mean ± SEM by two-way repeated measure ANOVA with Tukey post hoc method. **b** Proportions of time in each vigilance state during light and dark periods. **c** Bout count and **d** bout length for each vigilance state during light and dark periods. Data are presented as mean ± SEM. *$P < 0.05$ by unpaired *t*-test.

## Constitutive activation of AMPK in *Crbn*[−/−] mice during SD

As AMPK activation contributes to sleep homeostasis by regulating low-frequency delta power[3,8], we examined AMPK activity during SD. Brains of sleep-deprived mice were collected at different times during SD and analyzed by western blot (Fig. 4a, b). AMPK activation as reflected by phosphorylation of the AMPKα subunit increased during SD in both *Crbn*[+/+] and *Crbn*[−/−] mice. Interestingly, *Crbn*[−/−] mice showed higher levels of phosphorylated AMPKα in the non-sleep-deprived state compared with *Crbn*[+/+] mice, indicating that AMPK is constitutively more activated in *Crbn*[−/−] mice than in *Crbn*[+/+] mice. By contrast, levels of AMPKα and AMPKβ were not significantly altered during SD. The AMPKγ subunit was highly expressed in *Crbn*[−/−] mice, as previously reported[12], and further increased at the end of SD. However, levels of CAMKKII, the upstream kinase of AMPK, were constant during SD, suggesting that AMPK activation during SD is not attributed to the upregulation of the upstream kinase. To determine if this constitutive activation of AMPK caused the increase in chaperone proteins, We treated Compound C (CC) to inhibit AMPK activation before SD and examined whether SD-induced increase in chaperone proteins and less phosphorylation of Tau and α-Syn in *Crbn*[−/−] mice compared with *Crbn*[+/+] mice was suppressed. CC significantly suppressed the activation of AMPK after SD in both *Crbn*[+/+] and *Crbn*[−/−] mice, without change in the expression level of each AMPK subunit AMPKα, AMPKβ, and AMPKγ (Fig. 4c, d). Also, CC-treated SD-*Crbn*[−/−] mice showed more increased p-Tau level than SD-*Crbn*[−/−] mice, however, the level of p-α-Syn and chaperone protein was not affected by CC treatment (Fig. 4e, f).

## Higher sleep continuity and low-delta power in *Crbn*[−/−] mice during recovery sleep (RS)

We next measured the proportions of wakefulness, NREM sleep, and REM sleep during a 6-h RS period after SD. During RS, *Crbn*[−/−] mice showed significantly increased wakefulness and less NREM sleep than *Crbn*[+/+] mice. Also, both genotypes of mice showed decreased wakefulness and increased NREM sleep during RS compared to baseline (BL) (Fig. 5a). However, *Crbn*[−/−] mice exhibited reduced number of NREM and REM bouts compared to *Crbn*[+/+] mice (Fig. 5b). In the case of bout length, we observed longer bout length of NREM sleep and REM sleep in *Crbn*[−/−] mice than

*Crbn*[+/+] mice, and decreased wake bout length in both genotypes of mice in RS (Fig. 5c). Thus, the reduced bout count of NREM sleep and REM sleep, coupled with increased bout length of NREM sleep and REM sleep, collectively imply an increased sleep continuity in *Crbn*[−/−] mice. No difference in normalized NREM delta power (0.5–4.0 Hz) between genotypes was observed during RS. However, *Crbn*[−/−] mice showed higher delta power in the lower frequency band (0.5–2.5 Hz) during the first 4 h of the RS period (Fig. 5d, e).

## Discussion

In this study, overall sleep patterns were similar between genotypes during spontaneous sleep–wake cycles. However, slow-wave activity at the low-delta range (0.5–2.5 Hz) was significantly higher during RS in *Crbn*[−/−] mice. While SD-induced increase of p-AMPK was more enhanced in *Crbn*[−/−] mice, expression of neurodegenerative proteins, including p-Tau and p-α-synuclein, were attenuated with higher levels of chaperon proteins, including HSP70/DJ2 and DJ1. The evidence from our data indicates that *Crbn*[−/−] mice exhibit an augmented homeostatic sleep response and demonstrate a greater protective effect against neurodegenerative processes in the context of SD. To our knowledge, this study reports on the role of CRBN in modulating sleep–wake patterns and neurodegenerative processes.

In *Crbn*[+/+] mice, SD downregulated CRBN protein without affecting other substrate receptors of CRL4, namely DCAF1 and DCAF2. Prior research indicates that the abnormal sleep–wake cycle hastens the aggregation and increases the hyperphosphorylation of neuronal proteins such as Tau and α-synuclein, which promotes polymerization and fibrilization of each other[26–28]. Tau protein undergoes various post-translational modifications such as acetylation, glycosylation, ubiquitination, and phosphorylation. Among these modifications, phosphorylation is one of the earliest and most common modifications linked to the development of pathological inclusions. In the various tau isoforms, there are commonly more than 80 potential phosphorylation sites, which include serine, threonine, and tyrosine residues. Specifically, the phosphorylation of tau protein at various sites including Ser202/Thr205 induces aggregation, which is known to cause tauopathies. In the present study, we found that SD increased levels of the

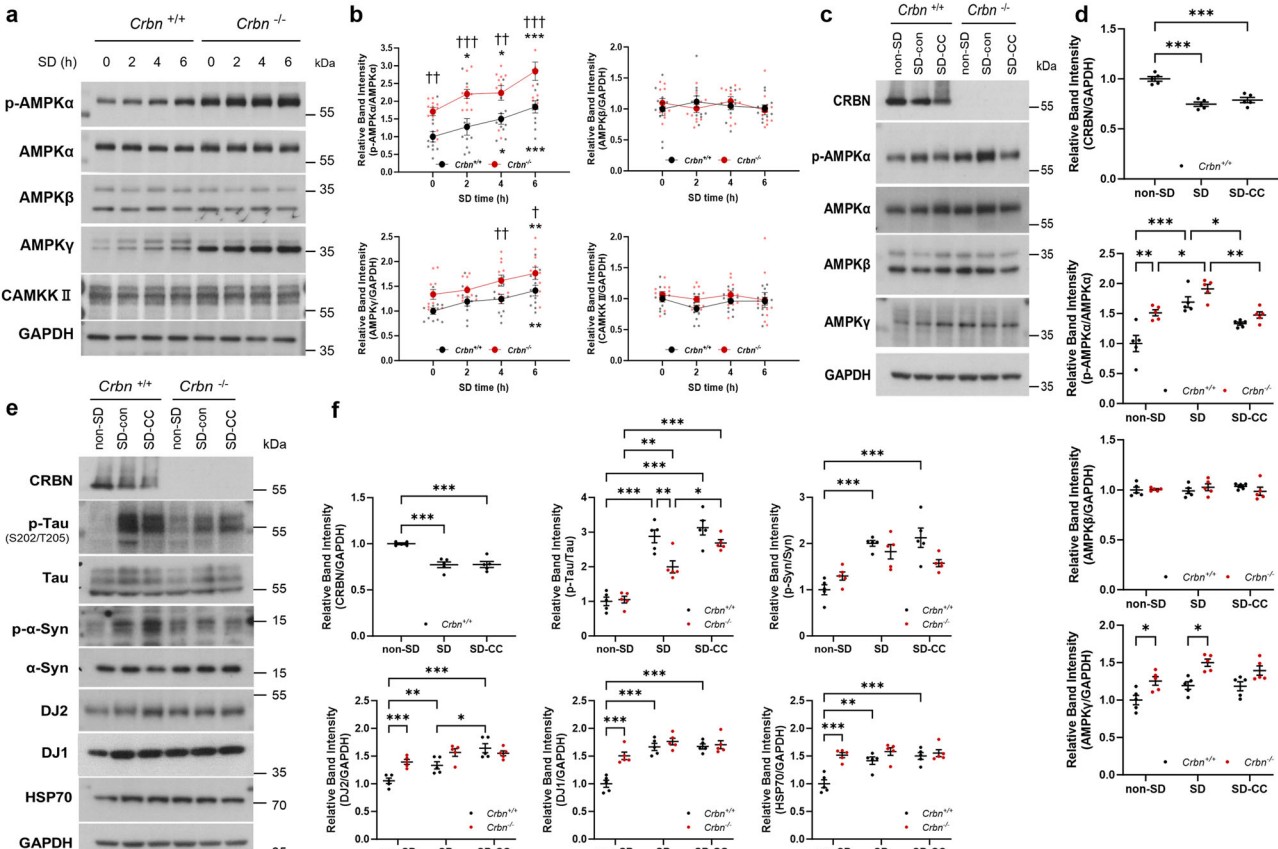

**Fig. 4 | Constitutive activation of AMPK in *Crbn*$^{-/-}$ mice during SD.** Mouse brains (n = 11 at each time point) were collected at different times during SD. **a** Protein levels of each AMPK subunit and its upstream regulator, CAMKKII, were measured by western blot analysis. **b** Band intensity of phosphorylated AMPKα (p-AMPKα), AMPKβ, AMPKγ, and CAMKKII relative to GAPDH or AMPKα was calculated by densitometric analysis. The image is representative of independent experiments, and data are presented as mean ± SEM by two-way repeated measure ANOVA with the Tukey post hoc method. The P values were presented as $^*P < 0.05$,

$^{**}P < 0.01$, $^{***}P < 0.001$ vs mice of the same genotype at 0 h. $^†P < 0.05$, $^{††}P < 0.01$, $^{†††}P < 0.001$ vs *Crbn*$^{+/+}$ mice at same time point. Brains of CC-injected WT and CRBN KO mice (n = 5 for each group) were collected after SD and the thalamus was isolated to make lysates. **c, d** Protein levels of the AMPK subunit were measured, and **e, f** each neuropathological protein and chaperone protein were measured by western blot. The image is representative of independent experiments, and data are presented as mean ± SEM by two-way repeated measure ANOVA with the Tukey post hoc method. The P values were presented as $^*P < 0.05$, $^{**}P < 0.01$, $^{***}P < 0.001$.

chaperone HSP70 and co-chaperones DJ2 and DJ1. Our research group previously reported that CRBN negatively regulates the activity of HSP70/DJ2 via ubiquitin-dependent proteasomal degradation[25]. As HSP70/DJ2 and DJ1 reduce the aggregation of Tau and α-synuclein, respectively[29,30], our findings suggest that the downregulation of CRBN by SD increases the levels of these chaperone proteins and thereby counteracts the aggregation of Tau and α-synuclein.

In spontaneous sleep–wake cycles, the phosphorylation of Tau and α-synuclein in *Crbn*$^{-/-}$ mice was lower than that of Crbn$^{+/+}$ mice, possibly due to higher expression of HSP70/DJ2 and DJ1. Additionally, after SD, *Crbn*$^{-/-}$ mice showed a lesser extent of increase in the phosphorylation of Tau and α-synuclein than *Crbn*$^{+/+}$ mice did, again likely due to the increased expression of chaperone proteins, such as HSP70/DJ2 and DJ1. These chaperone proteins play a crucial role in mitigating the aggregation of neurodegenerative proteins, as evidenced by the decreased levels of p-Tau and phosphorylated α-synuclein (p-α-Syn) observed in *Crbn*$^{-/-}$ mice. Previous studies have shown that HSP70 and DJ1 can inhibit the aggregation of Tau and α-synuclein, respectively, by promoting their proper folding and degradation[31,32]. These findings suggest that CRBN deficiency protects against the effects of SD by reducing the aggregation of Tau/α-synuclein and inducing high expression of HSP70/DJ2 and DJ1 chaperones.

*Crbn*$^{-/-}$ mice also showed increased levels of phosphorylated AMPK and constitutive activation of AMPK during SD, which could result in increased ATP synthesis and may be attributed to an altered CRBN level rather than the activity of CAMKKII, the upstream kinase of

AMPK. ATP has a dual function in regulating the sleep–wake cycle; it can serve as an excitatory neurotransmitter that promotes wakefulness via neuronal P2X purine receptors[33] but can also promote sleep via degradation into adenosine or stimulation of astrocytes to release interleukin-1β and tumor necrosis factor-α through astrocytic P2X receptors[29,30]. The increased AMPK activity in *Crbn*$^{-/-}$ mice could therefore lead to a more efficient restoration of energy balance during RS, contributing to the observed increased slow-wave activity and reduced neurodegenerative protein aggregation.

Interestingly, *Crbn*$^{-/-}$ mice exhibited reduced NREM sleep and increased EEG power in the low delta range during RS. This finding is reminiscent of human short sleepers who exhibit shorter sleep duration and increased slow wave activity in their EEG recordings[34]. Our data also demonstrated that the lack of *Crbn* suppressed the neurodegenerative processes, including p-Tau and synuclein. It is worth noting that the *DEC2* gene mutation, known to promote familial natural short sleep in humans, has been associated with increased wakefulness due to enhanced orexin expression[35,36]. Additionally, the DEC2 mutation has been shown to suppress neurodegenerative disorders, including AD[37]. The resemblance between the sleep patterns of *Crbn*$^{-/-}$ mice and short sleepers in humans suggests that the altered sleep pattern in *Crbn*$^{-/-}$ mice may reflect a beneficial sleep quality similar to that of short sleepers. Moreover, our findings highlight the potential connection between sleep patterns, neurodegeneration, and the role of the *DEC2* gene. Further research is warranted to explore the underlying mechanisms and potential therapeutic implications of these

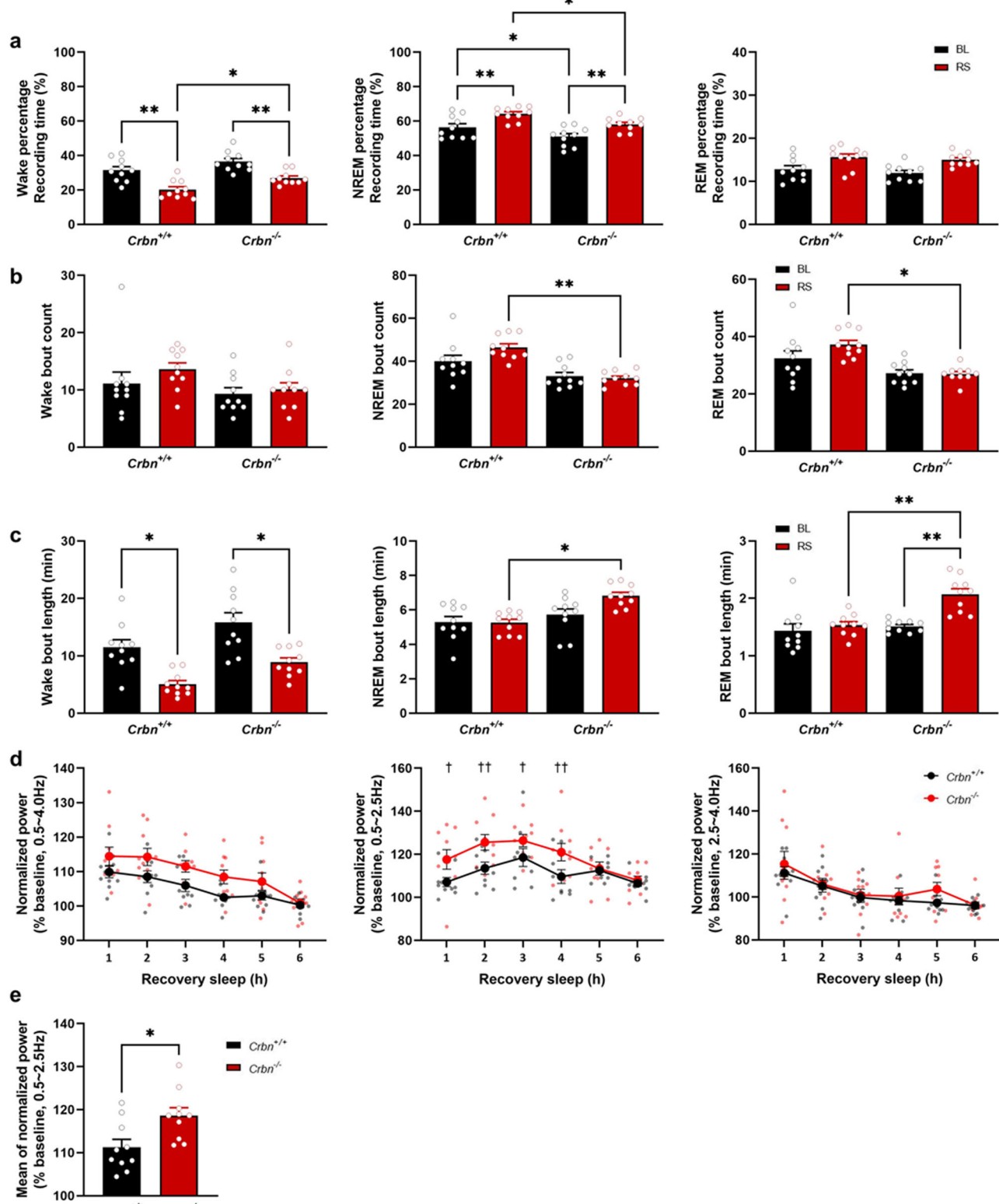

**Fig. 5 | Higher sleep continuity and NREM low-delta power in *Crbn⁻/⁻* mice during RS.** After SD, EEG recordings from *Crbn⁺/⁺* ($n = 10$) and *Crbn⁻/⁻* ($n = 10$) mice were obtained during a 6-h RS period. **a–c** Proportion of recording time, bout count, and length for each vigilance state. BL baseline, RS recovery sleep. Data are presented as mean ± SEM. $^*P < 0.05$, $^{**}P < 0.01$, $^{***}P < 0.001$ by two-way repeated measure ANOVA with Tukey post hoc method. **d** EEG delta power in the NREM state during RS was normalized by BL. Data were calculated by the power density (power/min) in each NREM sleep during RS as a percentage of NREM sleep during BL. Each frequency band expresses EEG delta power (0.5–4.0 Hz), low-delta power (0.5–2.5 Hz), and high-delta power (2.5–4.0 Hz). Data are presented as mean ± SEM, $^†P < 0.05$, $^{††}P < 0.01$ compared with *Crbn⁺/⁺* group by two-way repeated measure ANOVA with Tukey post hoc method. **e** Mean values of normalized NREM low-delta power across the entire 6-h RS period. Data are presented as mean ± SEM. $^{***}P < 0.001$ by unpaired *t*-test.

findings in improving sleep quality and mitigating neurodegenerative disorders.

To explore whether the observed effects in *Crbn*[−/−] mice were directly due to the absence of CRBN and not secondary effects, we attempted to use TD165 to knock down CRBN in vivo. While TD165 effectively knocked down CRBN in HEK293 cells, our in vivo experiments encountered significant challenges. Despite performing intracerebroventricular and intraperitoneal (IP) injections of TD165, we did not achieve CRBN knockdown in brain tissue. The high plasma protein binding of TD165 likely contributed to this issue[38]. Given the technical limitations in achieving CRBN knockdown in vivo with TD165, we recognize the need for alternative approaches to validate our findings. Future studies may explore other methods, such as gene editing or alternative knockdown techniques, to circumvent these challenges.

Thalidomide's teratogenic effects are related to CRBN, while its hypnotic effects are independent of the CRBN-mediated ubiquitin/proteasome pathway[39]. Given that our research aims to elucidate the molecular mechanisms by which CRBN deficiency affects sleep and neurodegeneration, investigating the hypnotic effect of thalidomide falls outside the scope of our study. While the hypnotic effects of thalidomide are an important area of research, our study is specifically designed to explore the pathways and mechanisms through which CRBN influences sleep and neurodegeneration.

The elevated ATP levels may contribute to increased alertness during RS, while the subsequent rise in extracellular adenosine, resulting from ATP degradation, may enhance homeostatic sleep pressure, evidenced by increased EEG power in the low-delta frequency range[30]. These results indicate that *Crbn*[−/−] mice exhibit a higher sleep drive when awake, making them more susceptible to sleepiness during SD. In essence, the absence of CRBN has the advantage of reduced neurodegeneration but may lead to a trade-off of reduced resistance to sleepiness. Consequently, it is plausible that CRBN has been preserved throughout evolution at the cost of increased risk for neurodegeneration, as it helps to moderate homeostatic sleep pressure by modulating AMPK levels, especially under stressful conditions like SD. Accordingly, we suggest that CRBN may serve as a potential target for enhancing homeostatic sleep pressure to treat insomnia and decreasing pathologic proteins to prevent neurodegenerative diseases, such as Alzheimer's disease and Parkinson's disease. Further investigation is warranted.

## Methods
### Experimental animals
Male *Crbn*[−/−] mice[13] and control C57BL/6J mice (*Crbn*[+/+] mice) were used at 3–6 months of age. Mice were housed in temperature (23 ± 2 °C) and humidity (55 ± 5%) controlled rooms under a 12:12-h light–dark cycle (lights on at 7 a.m.) with a standard chow diet and sterilized water ad libitum. All procedures were approved by the ethics committee of the Gwangju Institute of Science and Technology (GIST-2021-112) and fulfilled the Association for Assessment and Accreditation of Laboratory Animal Care International guidelines. We have complied with all relevant ethical regulations for animal use.

### Drugs
CC (Sigma-Aldrich, Cat No. 171260) was diluted in corn oil and injected intraperitoneally at 0.01 ml/g body weight of mouse for a final concentration of 10 mg/kg 1 h before SD.

### Surgical preparation, electrode implantation, and EEG recording
Mice were anesthetized with isoflurane (4% for induction, 1–3% for surgery) and mounted in a stereotaxic frame. As an analgesic, ketoprofen (5 mg kg$^{-1}$) was administered subcutaneously before surgery, and core body temperature was maintained with a heating pad. Bilateral holes were drilled in the skull for insertion of stainless steel electrodes (Pinnacle Technology, Inc., Lawrence, KS) for frontal EEG1 (relative to bregma: AP = 1.0 mm and ML = 1.0 mm) and parietal EEG2 (AP = −3.5 mm and ML = 1.0 mm) electrodes. An additional screw over the cerebellum (AP = 0.0 mm and ML = −5.3 mm) served as a reference. Two stainless steel wires were implanted in the nuchal muscle for electromyography (EMG) measurement. Mice were allowed to recover in their home cages for 7 days before EEG recordings.

After the mice recovered, they were placed in a soundproof chamber and attached to a swivel. After connecting the mice, they were habituated to the chamber for 24 h before EEG recording. EEG was recorded continuously for 24 h using the Sirenia Acquisition recording system (Pinnacle Technology). Acquired data were analyzed using Sirenia Sleep software. The EEG signals were sampled at a 2000 Hz sampling rate. Only artifact-free epochs were included in the frequency analysis.

### SD
Mice were sleep-deprived for 6 h (9 a.m. to 3 p.m.) by direct gentle handling using a soft brush and presenting novel objects[40,41]. During SD, mice were given access to food and water ad libitum.

### Twenty-four-hour EEG recording
EEG supported by video monitoring was recorded using Sirenia Acquisition software for 24 h to provide a BL. EEG and EMG recordings were divided into 10-s epochs, and manually labeled using Sirenia Sleep software. Wakefulness was identified by low-amplitude EEG oscillations of varying frequencies >4 Hz, typically accompanied by elevated EMG activity with phasic bursts. NREM sleep was determined by low-frequency, high-amplitude EEG oscillations heavy in slow delta wave activity with reduced EMG activity. REM sleep was identified by low-frequency, low-amplitude EEG oscillations, high theta wave activity, and an absence of EMG activity. Epochs that included more than one vigilance state were categorized based on which state accounted for >50% of the epoch. The proportions of time spent in wakefulness, NREM sleep, and REM sleep were calculated. The number and lengths of bouts, defined as three consecutive epochs with the same sleep or wake state, were also calculated. Average absolute power values were calculated for low-delta (0–2.5 Hz) and high-delta (2.5–4 Hz) frequency bands.

### Western blot analysis
For sample preparation, brains were isolated from mice and homogenized in cold Tris buffer (20 mM Tris-HCl, pH 7.4, 0.32 M sucrose, 1 mM EDTA, 1 mM EGTA, 1 mM PMSF, 10 mM aprotinin, 15 mM leupeptin, 50 mM NaF, and 1 mM Na$_3$VO$_4$) using a homogenizer (Thomas Scientific), and cleared by centrifugation at 12,500 rpm for 30 min. Brain lysates were quantified by Bradford protein assay (Bio-Rad) and boiled with 2× sample buffer (24 mM Tris-HCl, pH 6.8, 10% glycerol, 0.8% SDS, 0.04% bromophenol blue) for western blot analysis. Equal amounts of protein samples were separated through SDS-PAGE gel and transferred to polyvinylidene fluoride membranes using a Trans-Blot TurboTransfer System (Bio-Rad). Membranes were blocked in 4% bovine serum albumin (BSA) solution prepared in 1× TBS-T (16.5 mM Tris-HCl, pH 7.5, 137 mM NaCl, 2 mM KCl, 0.2% Tween 20) for 50 min at room temperature. Membranes were then incubated with specific primary antibodies: CRBN (CST, 1:1000), p-Tau (Thermo Fisher Scientific, 1:1000), p-Tau (CST, 1:1000), Tau (Abcam, 1:1000), p-α-synuclein (Abcam, 1:1000), α-synuclein (Abcam, 1:1000), DJ2 (Abcam, 1:1000), DJ1 (Abcam, 1:1000), HSP70 (CST, 1:1000), GAPDH (Abfrontier, 1:1000), p-AMPKα (CST, 1:1000), AMPKα (Invitrogen, 1:1000), AMPKβ (CST, 1:1000), AMPKγ (Abcam, 1:1000), CAMKKII (Invitrogen, 1:1000), DCAF1 (Invitrogen, 1:1000), DCAF2 (Abcam, 1:1000) (Detailed catalog number can be found in Supplementary Table 1) diluted in TBS-T at 4 °C. After three washes in TBS-T for 10 min, membranes were incubated in horseradish peroxidase-conjugated anti-rabbit or anti-mouse IgG (Jackson ImmunoResearch Laboratories, 1:10,000) as secondary antibodies for 50 min at room temperature. After three washes in TBS-T for 10 min, membranes were developed using enhanced chemiluminescence western blotting detection reagents (GE Healthcare Life Sciences, #RPN2209).

## Quantitative real-time PCR analysis

Total RNA was isolated from mouse brains by Hybrid-R (GeneAll) according to the manufacturer's protocol. Complementary DNA (cDNA) was synthesized from a 2 µg total RNA sample using MMLV Reverse Transcriptase (Thermo Fisher Scientific) according to the manufacturer's instructions. mRNA levels were measured by quantitative real-time polymerase chain reaction (PCR) using SYBR Select Master Mix (Thermo Fisher Scientific) and a StepOne Plus real-time PCR System (Applied Biosystems). The following primers were used for amplification: *Crbn*, forward 5'-GAT CCT GAT TCC TGG GCA GA-3' and reverse 5'-CAA GGA CTG CAA AGG TCC TG -3'; *Dcaf1*, forward 5'-AGC CTC TTC TCA TTG GCA CT-3' and reverse 5'-CAA GGA CTG CAA AGG TCC TG -3'; *Dcaf2*, forward 5'-GTG GGT TGG ACC TCT CAG AA-3' and reverse 5'-TAC AAG CTG CTG ACG AAG GA-3'; and *Gapdh*, forward 5'-CAT CAC TGC CAC CCA GAA GAC TG-3' and reverse 5'-TGG TGG ACC ACG AGT CAC ATC-3'. Protein expression was normalized to that of GAPDH.

## Immunohistochemistry

Mouse brains were collected and fixed in 4% paraformaldehyde at 4 °C. For cryoprotection, fixed brains were immersed in filtered 15% sucrose solution in phosphate-buffered saline (PBS) followed by 30% sucrose solution until brains sank, after which they were embedded in optimal cutting temperature (OCT) compound (Sakura Finetek). Samples in the OCT compound were sectioned along the coronal plane at 10 µm thickness, permeabilized by 0.5% Triton X-100 in PBS, and blocked in blocking buffer (1% BSA and 2% goat serum in PBS-T) for 1 h. Sectioned brain slices on slide glass were incubated with primary antibodies: CRBN (Abnova, 1:500), p-Tau (Thermo Fisher Scientific, 1:500), p-α-synuclein (Abcam, 1:500), DJ2 (Abcam, 1:500), DJ1 (Abcam, 1:500) overnight at 4 °C. After a brief wash, samples were incubated with Alexa Fluor-488 anti-rabbit and anti-mouse secondary antibodies (Thermo Fisher Scientific, 4 µg/ml) (Detailed catalog number can be found in Supplementary Table 1) for 2 h. Samples were counterstained with Hoechst dye (Abcam) for nuclei staining before mounting. An Olympus Fluoview Viewer was used to detect and measure fluorescence.

## Statistics and reproducibility

Age-matched male $Crbn^{+/+}$ control mice and male $Crbn^{-/-}$ mice ($n = 15$ in each group) were used in all in vivo experiments. Data shown in the figures were generated by at least three independent experiments, and the number of mice or tissue samples is also given in the figure legends. Twenty-four-hour BL sleep and 6-h RS state proportions, bout lengths, and bout counts were constructed by exporting sleep recording data from Sirenia into Microsoft Excel. Statistical analyses were conducted utilizing the Origin software package (OriginLab Corporation, Northampton, MA, USA) and Prism 9 (GraphPad Software, Inc., La Jolla, CA, USA). All data points were represented as mean ± standard error of the mean (SEM). The normality of the data distribution was assessed using the Shapiro-Wilk test. In cases where two groups were compared, the independent sample *t*-test was employed for datasets exhibiting normal distribution, while the non-parametric Mann–Whitney test was utilized for those not conforming to normal distribution. For comparisons involving three or more groups, a two-way analysis of variance (ANOVA) was implemented, followed by post-hoc Tukey's test to ascertain significant differences between group means. The threshold for statistical significance was set at a *P* value of less than 0.05 ($P < 0.05$).

## Data availability

Uncropped film data from the western blot can be found in Supplementary Fig. 4 in Supplementary Information. The source data behind the graphs in the paper can be found in Supplementary Data 1. Data are available upon a reasonable request.

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

## Acknowledgements

This work was supported by a Gwangju Institute of Science and Technology (GIST) Research Institute grant, funded by the GIST, South Korea; a Mid-career Research grant (2022R1A2C1005532); and the Original Technology Research Program for Brain Science (grant NRF-2018M3C7A1056293) of the National Research Foundation of Korea. This work was also funded by the Korean government, Ministry of Science and ICT (NRF-2022R1A2C3009749) and Ministry of Health & Welfare (HU22C0150).

## Author contributions

C.-S.P. and T.K. designed and supervised this research; J.-H.J., J. Kim, and U.A. performed experiments; J.-H.J., N.Y.L., J.-W.B., and S.J., contributed experimental animal care and technical support; J.-H.J., J. Kim, U.A., J.J., M.P., J. Kang, C.-S.P., and T.K. contributed unpublished reagents/analytic tools; J.-H.J., J. Kim, C.-S.P., and T.K. analyzed data; and J.-H.J. and J. Kim wrote the manuscript.

## Competing interests

The authors declare no competing interests.
