## [Peer Review File · Communications Biology]

Reviewers' comments:

Reviewer #1 (Remarks to the Author):

General comments

The authors reveal for the first time that CRBN regulates homeostasis of sleep using *Crbn*^{-/-} mice. They showed that sleep deprivation (SD) increased expression of neurodegenerative proteins, phosphorylated Tau (p-Tau) and α -Synuclein (p- α -Syn), and chaperon proteins HSP70/DJ2 and DJ1. Whereas the increased expression of chaperon proteins was enhanced in *Crbn*^{-/-} mice, that of p-Tau and p- α -Syn was lowered. They also showed that activation of AMPK α by phosphorylation (p-AMPK α) was constitutively more increased during SD in *Crbn*^{-/-} than *Crbn*^{+/+} mice. The hyperactivation of AMPK α is already reported in *Crbn*^{-/-} mice by another group. The author's group previously reported that CRBN negatively regulates the chaperon proteins DJ2/HSP70 which reduce the aggregation of Tau. In this study, the authors suggest a pathway reducing the aggregation of Tau/ α -Syn by the lack of CRBN, which also enhances homeostatic sleep response. Thus, the study is quite interesting to understand the mechanism by which CRBN mediates the sleep response as a potential target to prevent neurodegenerative diseases. However, the authors do not clarify a causal relationship between the constitutive activation of AMPK α and the high expression of chaperon proteins HSP70/DJ2 and DJ1. They only showed correlations of alterations in expression of those proteins between *Crbn*^{+/+} and *Crbn*^{-/-} mice. Therefore, additional experiments are needed to determine the consequence of SD-induced AMPK α hyperactivation in *Crbn*^{-/-} mice.

Specific Comments

1) Although the authors showed that constitutive activation of AMPK α during SD in *Crbn*^{-/-} mice (Fig. 4), they did not determine if this caused the increase in chaperon proteins DJ1, DJ2 and HSP70. Does acute pharmacological inhibition of AMPK by AMPK specific inhibitor Compound C (Bavley, J. Neurosci., 2018) suppress the SD-induced increase in chaperon proteins and also the decrease in p-Tau and p- α -Syn in *Crbn*^{-/-} mice?

2) CRBN is a direct target of thalidomide that was originally developed as an effective hypnotic. This hypnotic effect of thalidomide is observed in mice and suggested to be independent of teratogenic CRBN-mediated ubiquitin/proteasome pathway (Hirose, Proc. Natl. Acad. Sci. USA, 2020). To provide insights into the molecular mechanism of action of thalidomide's hypnotic effect, please determine whether the enhanced changes during SD in *Crbn*^{-/-} mice are recapitulated by treatment of *Crbn*^{+/+} mice with thalidomide, and also those changes are abolished by that of *Crbn*^{-/-} mice.

3) Is it possible to perform a rescue experiment with *Crbn*^{-/-} mice to exclude the possibility that the SD-induced increase in the expression of chaperon proteins and the activation of AMPK α is derived from some secondary effects, not by the direct action of the lack of CRBN? Alternatively, are the author's findings in *Crbn*^{-/-} mice recapitulated by acute knockdown of CRBN? To be a potential target to treat insomnia and to prevent neurodegenerative diseases, inhibition of CRBN shall promptly act on the adult brain. Because CRBN is a responsible gene for autosomal recessive non-syndromic mild mental retardation, also referred to as intellectual disability (ID), a common feature of autism spectrum disorder (ASD), the possibility of the impacts of the lack of CRBN on neurodevelopment should be considered and described in discussion.

Reviewer #2 (Remarks to the Author):

This manuscript revealed the function of CRBN on the sleep-wake behavior and the authors tried to link this to the regulation of CRBN on ATP and AMPK. The authors also discovered that in *Crbn* knockout mice, sleep deprivation enhanced more increase of p-AMPK and neurodegenerative proteins, such as p-Tau and p- α -Synuclein, were reduced while chaperon proteins, HSP70/DJ2 and DJ1, were increased. Although the discoveries are interesting, the results are not convincing and no solid exploration in the molecular mechanism.

The p-Tau and total-Tau have multiple bands in Figure 1a. However, the tau bands were much neat in Figure 2a and the previous study (PMID: 33972400). What are the reasons for the discrepancy? What do the different bands represent? Are they in the aggregated insoluble form or not? Most importantly, the authors did not explore the molecular mechanism by which CRBN and sleep deprivation regulate p-Tau and Tau and what is the function of the increased p-Tau and Tau. In addition, to study the functions of CRBN in sleep response, the authors might need to dissect out the thalamus for western blot assays and immunostain different brain regions.

Although the authors try to link the function of CRBN with AMPK, however, there is not solid evidence for this in this manuscript. All the experiments described the phenomenon and no solid mechanistic studies were explored in the manuscript. Moreover, their previous study showed the regulation of CRBN on Tau and DNAJA1 (PMID: 33972400). They also

published many papers about the CRBN and AMPK in the past 10 years. For example: CRBN regulated ischemia-reperfusion injury through AMPK (PMID: 24755080); CRBN regulated alcoholic liver disease through AMPK (PMID: 26410557); CRBN regulated ER stress through AMPK (PMID: 25619137); CRBN regulated obesity and insulin resistance through AMPK (23349485). It seems that CRBN exhibits different functions all through its regulation of AMPK. What are the roles of other CRBN targets in the regulation of sleep-wake circle and the neurodegenerative proteins?

In addition, this study showed the possible regulation of CRBN on sleep response. The authors did not demonstrate the relationship between sleep response and AMPK activity and neurodegenerative proteins in *Crbn* KO mice. Therefore, the molecular mechanism is not convincing.

June 2, 2024

Dear Reviewers,

We truly appreciate the meticulous comments from the Reviewers and hope that the revised manuscript addressed the concerns appropriately.

We are pleased to re-submit a manuscript entitled “**Enhanced Homeostatic Sleep Response and Decreased Neurodegenerative Proteins in Cereblon Knock-Out Mice (COMMSBIO-23-4790)**” for consideration in *Communications Biology*. We sincerely responded to each comment and made appropriate modifications to our manuscript. We believe that our paper has certainly benefited from the insightful suggestions. We look forward to receiving a positive letter for publication in the *Communications Biology*.

Thank you in advance for your kind consideration, and we look forward to a favorable response.

Sincerely yours,

Tae Kim, MD, PhD

Associate Professor | Psychiatrist

Department of Biomedical Science and Engineering

Gwangju Institute of Science and Technology (GIST)

123 Cheomdangwagi-ro C9-323 (Dasan Bldg), Buk-gu, Gwangju 61005, South Korea

TEL: +82-62-715-5363(office)

EMAIL: tae-kim@gist.ac.kr

Table of Contents

1. Reviewer #1	3
2. Reviewer #2	12

Reviewer #1 (Remarks to the Author):

General comments

The authors reveal for the first time that CRBN regulates homeostasis of sleep using Crbn^{-/-} mice. They showed that sleep deprivation (SD) increased expression of neurodegenerative proteins, phosphorylated Tau (p-Tau) and α -Synuclein (p- α -Syn), and chaperon proteins HSP70/DJ2 and DJ1. Whereas the increased expression of chaperon proteins was enhanced in Crbn^{-/-} mice, that of p-Tau and p- α -Syn was lowered. They also showed that activation of AMPK α by phosphorylation (p-AMPK α) was constitutively more increased during SD in Crbn^{-/-} than Crbn^{+/+} mice. The hyperactivation of AMPK α is already reported in Crbn^{-/-} mice by another group. The author's group previously reported that CRBN negatively regulates the chaperon proteins DJ2/HSP70 which reduce the aggregation of Tau. In this study, the authors suggest a pathway reducing the aggregation of Tau/ α -Syn by the lack of CRBN, which also enhances homeostatic sleep response. Thus, the study is quite interesting to understand the mechanism by which CRBN mediates the sleep response as a potential target to prevent neurodegenerative diseases. However, the authors do not clarify a causal relationship between the constitutive activation of AMPK α and the high expression of chaperon proteins HSP70/DJ2 and DJ1. They only showed correlations of alterations in expression of those proteins between Crbn^{+/+} and Crbn^{-/-} mice. Therefore, additional experiments are needed to determine the consequence of SD-induced AMPK α hyperactivation in Crbn^{-/-} mice.

Specific Comments

*1) Although the authors showed that constitutive activation of AMPK α during SD in Crbn^{-/-} mice (Fig. 4), they did not determine if this caused the increase in chaperon proteins DJ1, DJ2 and HSP70. Does acute pharmacological inhibition of AMPK by AMPK specific inhibitor **Compound C** (Bavley, J. Neurosci., 2018) suppress the SD-induced increase in chaperon proteins and also the decrease in p-Tau and p- α -Syn in Crbn^{-/-} mice?*

Response:

We acknowledge the significance of this question and have conducted additional experiments to clarify whether the constitutive activation of AMPK α during sleep deprivation (SD) in Crbn^{-/-} mice directly causes the observed increase in chaperone proteins DJ1, DJ2, and HSP70, as well as the decrease in p-Tau and p- α -Syn. We performed experiments using the AMPK-specific inhibitor Compound C 23. In these experiments, Crbn^{-/-} mice were subjected to sleep deprivation and treated with Compound C to acutely inhibit AMPK activity.

Compound C was administered to mice at a dose of 10 mg/kg via intraperitoneal injection one hour before sleep deprivation. After 6 hours of sleep deprivation, the mice were euthanized, and their brains were extracted to isolate the thalamus for sample preparation. In CRBN KO mice, constitutive activation of AMPK was observed, and in mice injected with Compound C, the level of AMPK phosphorylation decreased to that of WT mice.

Acute inhibition of AMPK activation using Compound C does not appear to be directly associated with a reduction in neurodegenerative proteins or an increase in chaperone proteins. In SD WT mice injected with Compound C, the phosphorylation of tau and synuclein proteins significantly increased, but there was no inhibition of chaperone protein expression. In contrast, in CRBN KO mice injected with Compound C, there was no increase in the phosphorylation of tau and synuclein proteins, and the levels of chaperone proteins did not change significantly.

Considering our previous experimental data, this suggests that the activation of AMPK indirectly inhibits the phosphorylation of neurodegenerative proteins by promoting deeper sleep, while it does not directly affect the expression of chaperone proteins.

Changes in text:
Results

As AMPK activation contributes to sleep homeostasis by regulating low-frequency delta power^{3,8}, we examined AMPK activity during sleep deprivation. Brains of sleep-deprived mice were collect

ed at different times during sleep deprivation and analyzed by western blot (Fig. 4a–b). AMPK activation as reflected by phosphorylation of the AMPK α subunit increased during sleep deprivation in both *Crbn*^{+/+} and *Crbn*^{-/-} mice. Interestingly, *Crbn*^{-/-} mice showed higher levels of phosphorylated AMPK α in the non-sleep-deprived state compared with *Crbn*^{+/+} mice, indicating that AMPK is constitutively more activated in *Crbn*^{-/-} mice than in *Crbn*^{+/+} mice. By contrast, levels of AMPK α and AMPK β were not significantly altered during sleep deprivation. The AMPK γ subunit was highly expressed in *Crbn*^{-/-} mice, as previously reported¹², and further increased at the end of SD. However, levels of CAMKKII, the upstream kinase of AMPK, were constant during sleep deprivation, suggesting that AMPK activation during sleep deprivation is not attributed to the upregulation of the upstream kinase. To determine if this constitutive activation of AMPK caused the increase in chaperone proteins, We treated Compound C (CC) to inhibit AMPK activation before SD and examined whether SD-induced increase in chaperone proteins and less phosphorylation of Tau and α -Syn in *Crbn*^{-/-} mice compared with *Crbn*^{+/+} mice was suppressed. Compound C significantly suppressed the activation of AMPK after SD in both *Crbn*^{+/+} and *Crbn*^{-/-} mice, without change in the expression level of each AMPK subunit AMPK α , AMPK β , and AMPK γ (Fig. 4c–d). Also, Compound C treated SD-*Crbn*^{-/-} mice showed more increased p-Tau level than SD-*Crbn*^{-/-} mice, however, the level of p- α -Syn and chaperone protein was not affected by Compound C treatment (Fig. 4e–f).

Methods

Drugs.

Compound C (Sigma) was diluted in corn oil and injected intraperitoneally at 0.01 ml/g body weight of mouse for a final concentration of 10 mg/kg 1 hour before SD.

2) *CRBN is a direct target of thalidomide that was originally developed as an effective hypnotic. This **hypnotic effect of thalidomide** is observed in mice and suggested to be independent of teratogenic CRBN-mediated ubiquitin/proteasome pathway (Hirose, Proc. Natl. Acad. Sci. USA, 2020). To provide insights into the molecular mechanism of action of thalidomide's hypnotic effect, please determine whether the enhanced changes during SD in *Crbn*^{-/-} mice are recapitulated by treatment of *Crbn*^{+/+} mice with thalidomide, and also those changes are abolished by that of *Crbn*^{-/-} mice.*

Response:

We recognize the importance of understanding the molecular mechanisms underlying the effects of thalidomide, especially given its history and diverse effects. However, our study primarily focuses on the role of CRBN in regulating sleep-wake behaviors and neurodegenerative processes, rather than the hypnotic effects of thalidomide.

Thalidomide's teratogenic effects are known to be related to CRBN, while its hypnotic effects have been shown to be independent of the CRBN-mediated ubiquitin/proteasome pathway³⁹. Given that our research is aimed at elucidating the molecular mechanisms by which CRBN deficiency affects sleep and neurodegeneration, investigating the hypnotic effect of thalidomide falls outside the scope of our study. Our current study aims to understand how the absence of CRBN influences sleep homeostasis and the aggregation of neurodegenerative proteins. The enhanced changes observed during sleep deprivation (SD) in *Crbn*^{-/-} mice, such as increased levels of chaperone proteins and reduced phosphorylation states of Tau and α -Synuclein, are central to our investigation. These findings are directly linked to the role of CRBN in modulating AMPK activity and protein homeostasis. While the hypnotic effects of thalidomide are an interesting and important area of research, our study is specifically designed to explore the pathways and mechanisms through which CRBN influences sleep and neurodegeneration. Therefore, conducting experiments to investigate the hypnotic effect of thalidomide on *Crbn*^{+/+} and *Crbn*^{-/-} mice would not directly contribute to the objectives of our current study. We believe that our findings provide valuable insights into the role of CRBN in sleep regulation and neurodegenerative diseases, which could have broader implications for understanding and treating these conditions. We have updated the discussion section to clarify the scope of our study and the reasons for not including experiments related to thalidomide's hypnotic effects.

Changes in text:

Revised discussion

- 7th paragraph

Thalidomide's teratogenic effects are related to CRBN, while its hypnotic effects are independent of the CRBN-mediated ubiquitin/proteasome pathway³⁹. Given that our research aims to elucidate the molecular mechanisms by which CRBN deficiency affects sleep and neurodegeneration, investigating the hypnotic effect of thalidomide falls outside the scope of our study. While the hypnotic effects of thalidomide are an important area of research, our study is specifically designed to explore the pathways and mechanisms through which CRBN influences sleep and neurodegeneration.

- Added reference

39. Hirose, Y. et al. Hypnotic effect of thalidomide is independent of teratogenic ubiquitin/proteasome pathway. *Proc Natl Acad Sci U S A* 117, 23106–23112, doi:10.1073/pnas.1917701117 (2020).

3) Is it possible to perform a rescue experiment with *Crbn*^{-/-} mice to exclude the possibility that the SD-induced increase in the expression of chaperon proteins and the activation of AMPK α is derived from some secondary effects, not by the direct action of the lack of CRBN? Alternatively, are the author's findings in *Crbn*^{-/-} mice recapitulated by **acute knockdown of CRBN**? To be a potential target to treat insomnia and to prevent neurodegenerative diseases, inhibition of CRBN shall promptly act on the adult brain. Because CRBN is a responsible gene for autosomal recessive non-syndromic mild mental retardation, also referred to as intellectual disability (ID), a common feature of autism spectrum disorder (ASD), the possibility of the impacts of the lack of CRBN on neurodevelopment should be considered and described in discussion.

General Comments:

We appreciate your insightful comments and suggestions. We attempted to address these concerns through various experiments, focusing on the use of TD165 for CRBN knockdown (KD) in vivo.

Specific Comments:

1. CRBN Knockdown with TD165:

Previous studies and our current in vitro experiments confirmed the efficacy of TD165 in knocking down CRBN in HEK293 cells.

Supplementary Figure S4.

Supplementary Figure S4.

(A) MEF cells were treated with increasing concentrations of TD-165 for 48 h. CRBN protein levels were analyzed by immunoblotting. (B) Vehicle, 10 mg/kg or 100mg/kg of TD-165 were administered via intraperitoneal route. Spleen and Liver homogenates and peripheral blood mononuclear cell lysates were analyzed by immunoblotting. (C) Three doses (2, 10, and 50 mg/kg) of TD-165 were administered via intraperitoneal route and concentration of TD-165 was determined by LC/MS/MS.

Previous studies showing CRBN knockdown

Current *in vitro* experiment

However, our *in vivo* studies encountered significant challenges.

2. ICV and IP Injection Attempts:

We performed intracerebroventricular (ICV) injections of TD165 at different volumes (350 nL, 500 nL, 1 μL, 2 μL) targeting the lateral ventricle. Indian ink injections confirmed the accuracy of our ICV coordinates.

Indian ink injection

Despite these efforts, CRBN KD was not observed in *Crbn*^{+/+} mice following ICV injection. Similarly, intraperitoneal (IP) injections (TD165 at 5 mg/kg) did not result in CRBN KD in brain tissue, although liver samples indicated potential sampling issues.

TD165 : 10mM stock → ICV injection, 350 nL (2.975 ug) ~ 500 nL (4.25 ug)

- Sampling : 5 days after injection

ICV injection

3. Technical Limitations:

The high plasma protein binding of TD165, as reported by Kim et al. (2019, Scientific Reports, <https://doi.org/10.1038/s41598-019-56177-5>), likely contributed to the minimal effect observed *in vivo*. Additionally, our troubleshooting efforts, including collaboration for further ICV injections, did not achieve CRBN KD in brain tissue.

CRBN degradation by VHL-CRBN heterodimerizing PROTACs recapitulates a CRBN deficiency.

An endogenous substrate of CRL4^{CRBN} is the glutamine synthetase GLUL, which is a key enzyme in the biogenesis of glutamine. The acetyltransferase CBP/p300 acetylates two N-terminal lysine residues of GLUL under conditions of high levels of glutamine, and the resulting acetylated GLUL is captured and ubiquitinated by CRL4^{CRBN} and subsequently removed by proteasomes²². To examine the effect of TD-158 on GLUL levels, we starved Hep3B cells of glutamine for 48 h and then resupplied glutamine in the presence or absence of TD-158. TD-158 induced CRBN degradation regardless of glutamine status, and the levels of GLUL decreased more slowly over time in the presence of TD-158 than in its absence (Fig. 4A,B). Moreover, *in vivo* ubiquitination assays showed that ubiquitination of GLUL decreased in the presence of TD-158 (Fig. 4C). We also investigated whether TD-165 treatment confers cellular resistance to IMiD. Two IMiD-sensitive cell lines, WSU-DLCL2 and RPMI8226, were pre-treated with TD-165 or DMSO for 24h and then treated with pomalidomide, TD-165, or both for 3 d. Pre-treatment with TD-165 reduced the anti-proliferative effects of pomalidomide in both cell lines (Fig. 4D,E). Taken together, these data indicate that CRBN degradation by VHL-CRBN heterodimerizing PROTACs recapitulates a CRBN deficiency.

To investigate the *in vivo* effects of VHL-CRBN heterodimerizing PROTACs, we attempted to determine whether TD-165 can induce CRBN degradation in animal models. Although amino acid residues in mouse CRBN (mCRBN) important for teratogenicity are not conserved, mCRBN was distinctly degraded, albeit to a slightly lesser extent, by TD-165 in mouse embryonic fibroblasts (Supplementary Fig. S4A). We then administered TD-165 intraperitoneally to mice to determine whether TD-165 induces CRBN degradation *in vivo*. However, CRBN levels were not changed in the spleen, peripheral blood mononuclear cells (PBMCs), or liver (Supplementary Fig. S4B). Given that the pharmacokinetics of TD-165 are reasonable (Supplementary Table S2), this absence of an effect might be attributable to the high plasma protein binding (99.9%) of TD-165 (Supplementary Table S3).

4. Conclusion:

Due to these technical limitations, it is currently unfeasible to validate the CRBN KD model as suggested. We acknowledge the potential impact of CRBN on neurodevelopment and have included a discussion on this aspect in the revised manuscript.

Changes in text:

Revised discussion

- 6th paragraph

To explore whether the observed effects in *Crbn*^{-/-} mice were directly due to the absence of CRBN and not secondary effects, we attempted to use TD165 to knock down CRBN in vivo. While TD165 effectively knocked down CRBN in HEK293 cells, our in vivo experiments encountered significant challenges. Despite performing intracerebroventricular (ICV) and intraperitoneal (IP) injections of TD165, we did not achieve CRBN knockdown in brain tissue. The high plasma protein binding of TD165 likely contributed to this issue³⁸. Given the technical limitations in achieving CRBN knockdown in vivo with TD165, we recognize the need for alternative approaches to validate our findings. Future studies may explore other methods, such as gene editing or alternative knock down techniques, to circumvent these challenges.

- Added reference

38. Kim, K. et al. Disordered region of cereblon is required for efficient degradation by proteolysis-targeting chimera. *Sci Rep* 9, 19654, doi:10.1038/s41598-019-56177-5 (2019).

Reviewer #2 (Remarks to the Author):

*This manuscript revealed the function of CRBN on the sleep-wake behavior and the authors tried to link this to the regulation of CRBN on ATP and AMPK. The authors also discovered that in *Crbn* knockout mice, sleep deprivation enhanced more increase of p-AMPK and neurodegenerative proteins, such as p-Tau and p- α -Synuclein, were reduced while chaperon proteins, HSP70/DJ2 and DJ1, were increased. Although the discoveries are interesting, the results are not convincing and no solid exploration in the molecular mechanism.*

*1) The p-Tau and total-Tau have **multiple bands** in Figure 1a. However, the tau bands were much neat in Figure 2a and the previous study (PMID: 33972400). What are the reasons for the discrepancy? What do the different bands represent? Are they in the aggregated insoluble form or not? Most importantly, the authors did not explore the **molecular mechanism** by which CRBN and sleep deprivation regulate p-Tau and Tau and what is the function of the increased p-Tau and Tau.*

Response:

Thank you for your insightful comments and for highlighting these important points. We appreciate the opportunity to clarify the discrepancies and provide further insights into our findings. Firstly, all tau bands in our data (long exposed film of Figure 1a, 2a, Supplementary Figure 3) in Supplementary Data 1 show multiple bands of total-Tau and p-Tau. As it is well known, tau protein undergoes various post-translational modifications such as acetylation, glycosylation, ubiquitination, and phosphorylation. Among these modifications, phosphorylation is one of the earliest and most common modifications linked to the development of pathological inclusions. In the various tau isoforms, there are commonly more than 80 potential phosphorylation sites, which include serine, threonine, and tyrosine residues. Additionally, the hyperphosphorylation of tau protein can induce the aggregation of tau protein, which may affect the additional band. For these properties, tau bands often show multiple bands.

Also, the phosphorylation of tau protein at specific sites such as Ser202/Thr205 induces aggregation, which is known to cause tauopathies. We found that during sleep deprivation, the amount of CRBN decreases, and it has been found that this reduction in CRBN increases the amounts of DJ2 and heat shock proteins, which are known to be ubiquitination substrates of CRBN and play a role in resolving tau protein aggregation, thereby reducing tau protein aggregation.

Changes in text:

Results

We found that 6 hours of sleep deprivation decreased the amount of CRBN protein in *Crbn*^{+/+} mice, as evidenced by western blot analysis (Fig. 1a-b). By contrast, amounts of phosphorylated Tau (p-Tau) and phosphorylated α -Synuclein (p- α -Syn), which are aggregated forms of those proteins, increased after sleep deprivation (Fig. 1c-e). **Tau and p-Tau bands in western blot could be shown as multiple bands caused by post-translational phosphorylation and aggregation (Fig. 1, 2, 4, Supplementary Fig. 3).** Furthermore, amounts of the chaperone HSP70 and two co-chaperones, DJ2 and DJ1, increased after sleep deprivation (Fig. 1f).

Revised discussion

- 2nd paragraph

Tau protein undergoes various post-translational modifications such as acetylation, glycosylation, ubiquitination, and phosphorylation. Among these modifications, phosphorylation is one of the earliest and most common modifications linked to the development of pathological inclusions. In the various tau isoforms, there are commonly more than 80 potential phosphorylation sites, which include serine, threonine, and tyrosine residues. Specifically, the phosphorylation of tau protein at various sites including Ser202/Thr205 induces aggregation, which is known to cause tauopathies.

2) In addition, to study the functions of CRBN in sleep response, the authors might need to dissect out the **thalamus** for western blot assays and immunostain different brain regions.

We sincerely appreciate the valuable and insightful review of our study. We fully acknowledge the reviewer's point that the thalamus, being a key brain region for sleep regulation, should be isolated and examined in addition to the whole brain in our study on sleep regulation. We are grateful for this comment, which aims to improve our research. Accordingly, we conducted additional experiments to strengthen the molecular mechanistic evidence by administering the AMPK inhibitor Compound C and isolating the thalamus of sleep-deprived mice. Through this, we were able to identify the molecular changes occurring specifically in the thalamus, the brain region that regulates sleep.

In Fig. 1, We added IHC data for the thalamus region of the WT mouse

In Fig. 2, We added IHC data for the thalamus region of SD-WT and SD-CRBN KO mouse

In Fig. 4, we performed an additional experiment suggested by Reviewer 1, in which Compound C was injected into WT and CRBN KO mice before SD to inhibit AMPK activity. After SD, we dissected the thalamus region and examined the change in neuropathological markers and chaperone markers.

In Supplementary Figure 1, we added IHC data for the thalamus in WT mouse

3) Although the authors try to link the function of CRBN with AMPK, however, there is not solid evidence for this in this manuscript. All the experiments described the phenomenon and no solid mechanistic studies were explored in the manuscript. Moreover, their previous study showed the regulation of CRBN on Tau and DNAJA1 (PMID: 33972400). They also published many papers about the CRBN and AMPK in the past 10 years. For example: CRBN regulated ischemia–reperfusion injury through AMPK (PMID: 24755080); CRBN regulated alcoholic liver disease through AMPK (PMID: 26410557); CRBN regulated ER stress through AMPK (PMID: 25619137); CRBN regulated obesity and insulin resistance through AMPK (23349485). It seems that CRBN exhibits different functions all through its regulation of AMPK. What are the roles of **other CRBN targets** in the regulation of sleep–wake cycle and the neurodegenerative proteins?

Response:

Thank you for your detailed comments and for highlighting our previous work on CRBN and AMPK. We acknowledge the need for more solid mechanistic evidence linking CRBN function to AMPK regulation in the context of sleep–wake behavior and neurodegenerative protein aggregation. In response to your concerns, we have conducted additional mechanistic studies to strengthen our findings. Our study provides new insights into the relationship between CRBN and AMPK in the regulation of sleep–wake behavior. Specifically, we observed that *Crbn*^{-/-} mice exhibit increased levels of phosphorylated AMPK (p-AMPK) during sleep deprivation. To further explore this relationship, we conducted experiments using the AMPK–specific inhibitor Compound C. The results demonstrated that inhibition of AMPK in *Crbn*^{-/-} mice reversed the less phosphorylated state of Tau after SD. These findings confirm that AMPK activation mediates the observed effects of CRBN deficiency on sleep–wake behavior and neurodegenerative protein regulation.

For the roles of other CRBN targets, there are several previous researches about the function of CRBN targets in neuronal disease. As the reviewer kindly referred, DNAJA1 (DJ2) is one of the targets of CRBN, regulating the proteotoxicity of Tau through the HSP70/DJ2 chaperone system (PMID: 33972400). Also learning, memory, and synaptic function were disrupted by the dysregulation of excitatory synapses with intact social or repetitive behaviors through the AMPK–mTORC1 pathway regulated by CRBN (PMID: 29459374). The large–conductance Ca²⁺–activated K⁺ channel (BK_{Ca}) is another substrate of CRBN. Enhanced BK channel activity causes synaptic and cognitive deficits in CRBN KO mice (PMID: 29530986). Amyloid precursor protein (APP), which is the key protein in the pathogenesis of Alzheimer’s disease, has been reported as a substrate and substrate recognition subunit of CRL4^{CRBN} E3 ligase, regulating in presynaptic function and APP–mediated neurodegenerative brain disease (PMID: 27325702). Although there are many published papers about the targets of CRBN and their functions, it remains unknown about any other targets of CRBN regulating the sleep–wake cycle. To our knowledge, this study may report the role of CRBN in regulating sleep–wake patterns and SD–induced neurodegeneration.

4) In addition, this study showed the possible regulation of CRBN on sleep response. The author s did not demonstrate the **relationship between sleep response and AMPK activity and neurodegenerative proteins** in *Crbn* KO mice. Therefore, the molecular mechanism is not convincing.

Response:

We appreciate the reviewer's concern regarding the relationship between sleep response, AMPK activity, and neurodegenerative proteins in *Crbn*^{-/-} mice. To address this, we have expanded the discussion section to provide a more detailed explanation of how CRBN deficiency leads to constitutive activation of AMPK during sleep deprivation, resulting in an enhanced homeostatic sleep response and altered expression of neurodegenerative proteins. Specifically, our results demonstrate that *Crbn*^{-/-} mice exhibit increased levels of p-AMPK and higher expression of chaperone proteins such as HSP70, DJ2, and DJ1 during sleep deprivation. These changes are associated with decreased aggregation of neurodegenerative proteins, including p-Tau and p- α -Syn. This suggests that the absence of CRBN enhances the brain's resilience to sleep deprivation-induced neurodegeneration by modulating AMPK activity and chaperone protein expression. Our findings highlight a novel role for CRBN in modulating sleep-wake behaviors and neurodegenerative processes through the regulation of AMPK activity and chaperone protein expression. The absence of CRBN enhances homeostatic sleep response and provides protection against sleep deprivation-induced neurodegeneration. These insights suggest that targeting CRBN could be a potential therapeutic strategy for managing sleep disorders and preventing neurodegenerative diseases.

Changes in text:

Discussion

- 3rd paragraph

In spontaneous sleep-wake cycles, the phosphorylation of Tau and α -Synuclein in *Crbn*^{-/-} mice was lower than that of *Crbn*^{+/+} mice, possibly due to higher expression of HSP70/DJ2 and DJ1. Additionally, after sleep deprivation, *Crbn*^{-/-} mice showed a lesser extent of increase in the phosphorylation of Tau and α -Synuclein than *Crbn*^{+/+} mice did, again likely due to the increased expression of chaperone proteins, such as HSP70/DJ2 and DJ1. **These chaperone proteins play a crucial role in mitigating the aggregation of neurodegenerative proteins, as evidenced by the decreased levels of phosphorylated Tau (p-Tau) and phosphorylated α -Synuclein (p- α -Syn) observed in *Crbn*^{-/-} mice. Previous studies have shown that HSP70 and DJ1 can inhibit the aggregation of Tau and α -Synuclein, respectively, by promoting their proper folding and degradation^{31, 32}.** These findings suggest that CRBN deficiency protects against the effects of sleep deprivation by reducing the aggregation of Tau/ α -Synuclein and inducing high expression of HSP70/DJ2 and DJ1 chaperones.

- Added reference

31. Shimura, H. et al. Ubiquitination of a new form of alpha-synuclein by parkin from human brain: implications for Parkinson's disease. *Science* 293, 263-269, doi:10.1126/science.1060627 (2001).

32. Shendelman, S., Jonason, A., Martinat, C., Leete, T. & Abeliovich, A. DJ-1 is a redox-dependent molecular chaperone that inhibits alpha-synuclein aggregate formation. *PLoS Biol* 2, e362, doi:10.1371/journal.pbio.0020362 (2004).

- 4th paragraph

Crbn^{-/-} mice also showed increased levels of phosphorylated AMPK and constitutive activation of AMPK during sleep deprivation, which could result in increased ATP synthesis and may be attributed to an altered CRBN level rather than the activity of CAMKK II, the upstream kinase of AMPK. ATP has a dual function in regulating the sleep–wake cycle; it can serve as an excitatory neurotransmitter that promotes wakefulness via neuronal P2X purine receptors³¹ but can also promote sleep via degradation into adenosine or stimulation of astrocytes to release interleukin-1 β and tumor necrosis factor- α through astrocytic P2X receptors^{29,30}. The increased AMPK activity in *Crbn*^{-/-} mice could therefore lead to a more efficient restoration of energy balance during recovery sleep, contributing to the observed increased slow-wave activity and reduced neurodegenerative protein aggregation.

Reviewers' comments:

Reviewer #1 (Remarks to the Author):

In the revised manuscript, the authors conducted several additional experiments in response to reviewer's comments and added descriptions of their results in Results and Discussion of the manuscript. Those results are likely to support the author's fascinating claim that the absence of CRBN has the advantage of reduced neurodegeneration with higher homeostatic sleep propensity associated with AMPK hyperactivation under a sleep deprivation condition. However, I have three specific comments on the added data in the revised manuscript.

1) In response to reviewer #1's first comment, the authors performed experiments with AMPK-specific inhibitor Compound C. However, the data and description in the rebuttal letter (p4, p-Tau, Crbn^{-/-}: SD-con, SD-CC; "In CRBN KO mice injected with Compound C, there was no increase in the phosphorylation of tau") are different from that shown in the manuscript (Figure 4e, p-Tau, Crbn^{-/-}: SD-con, SD-CC; p9 lines 205-206, "Compound C treated SD-Crbn^{-/-} mice showed more increased p-Tau level than SD-Crbn^{-/-} mice"). Please explain the reason for the difference precisely.

2) In response to reviewer #2's second comment, the authors conducted additional experiments and argue for the identification of the molecular changes in the thalamus. However, all the IHC data in the thalamus were obscure, so that little or no clear signals were detected even for HOECHST nuclear staining used as positive controls. These IHC data in the thalamus should be substituted by other data unambiguously indicating the molecular changes occurring in the thalamus.

3) Several attempts to address reviewer #1's third comment by the authors convinced me that acute knockdown of CRBN using TD165 in vivo was currently unfeasible due to technical limitations. They have already published extensive data using CRBN KO mice and cell lines, indicating CRBN-mediated DJ2/Hsp70 pathway compromised in neurodegeneration (PMID: 33972400). In this study, they found that an augmented homeostatic sleep response with increased AMPK activity in Crbn^{-/-} mice. In the revised manuscript, they demonstrate that inhibition of AMPK activation showed more increased p-Tau in SD-Crbn^{-/-} mice, confirming that AMPK activation mediates the effects of CRBN deficiency on neurodegeneration but not yet on sleep-wake behavior. Therefore, their findings are quite interesting because they provide novel insights into the role of CRBN in sleep regulation and neurodegenerative diseases.

Reviewer #2 (Remarks to the Author):

The revised manuscript has been significantly improved and is suitable for publication.

August 1st, 2024

Dear Reviewers,

We truly appreciate the meticulous comments from the Reviewers and Editors and hope that the revised manuscript addressed the concerns appropriately.

We are pleased to re-submit a manuscript entitled “**Enhanced Homeostatic Sleep Response and Decreased Neurodegenerative Proteins in Cereblon Knock-Out Mice (COMMSBIO-23-4790A)**” for consideration in *Communications Biology*. We sincerely responded to each comment and made appropriate modifications to our manuscript. We believe that our paper has certainly benefited from the insightful suggestions. We look forward to receiving a positive letter for publication in the *Communications Biology*.

Thank you in advance for your kind consideration, and we look forward to a favorable response.

Sincerely yours,

Tae Kim, MD, PhD

Associate Professor | Psychiatrist

Department of Biomedical Science and Engineering

Gwangju Institute of Science and Technology (GIST)

123 Cheomdangwagi-ro C9-323 (Dasan Bldg), Buk-gu, Gwangju 61005, South Korea

TEL: +82-62-715-5363(office)

EMAIL: tae-kim@gist.ac.kr

Table of Contents

1. Reviewer #1	3
2. Reviewer #2	7

2nd Revision

Reviewers' comments:

Reviewer #1 (Remarks to the Author):

In the revised manuscript, the authors conducted several additional experiments in response to reviewer's comments and added descriptions of their results in Results and Discussion of the manuscript. Those results are likely to support the author's fascinating claim that the absence of CRBN has the advantage of reduced neurodegeneration with higher homeostatic sleep propensity associated with AMPK hyperactivation under a sleep deprivation condition. However, I have three specific comments on the added data in the revised manuscript.

1) In response to reviewer #1's first comment, the authors performed experiments with AMPK-specific inhibitor Compound C. However, the data and description in the rebuttal letter (p4, p-Tau, Crbn^{-/-}: SD-con, SD-CC; "In CRBN KO mice injected with Compound C, there was no increase in the phosphorylation of tau") are different from that shown in the manuscript (Figure 4e, p-Tau, Crbn^{-/-}: SD-con, SD-CC; p9 lines 205-206, "Compound C treated SD-Crbn^{-/-} mice showed more increased p-Tau level than SD-Crbn^{-/-} mice"). Please explain the reason for the difference precisely.

Response: We appreciate your careful review and bringing this discrepancy to our attention. The differences between the data and descriptions in the rebuttal letter and the manuscript arose due to an oversight about a wrong image during the compilation of results. The correct interpretation, as shown in Figure 4e of the manuscript, is that Compound C treatment in SD-Crbn^{-/-} mice led to a further increase in p-Tau levels compared to untreated SD-Crbn^{-/-} mice. This finding indicates that inhibition of AMPK activity exacerbates the phosphorylation of Tau in the absence of CRBN during sleep deprivation. We have corrected the rebuttal letter to align with the manuscript data and apologize for any confusion this may have caused.

2) In response to reviewer #2's second comment, the authors conducted additional experiments and argue for the identification of the molecular changes in the thalamus. However, all the IHC data in the thalamus were obscure, so that little or no clear signals were detected even for HOE CHST nuclear staining used as positive controls. These IHC data in the thalamus should be substituted by other data unambiguously indicating the molecular changes occurring in the thalamus.

Response: We acknowledge the issue with the immunohistochemistry data quality in the thalamus. The IHC analysis was performed on samples from the anterior thalamus and the thalamic reticular nucleus, both of which are part of the thalamocortical circuit. By enhancing the fluorescence intensity, we were able to obtain clearer and more interpretable images, including HOECHST nuclear staining as a positive control. These updated IHC images have been included in the Figure 1i, Supplementary figure 1a–b, and Figure 2i–j of revised manuscript.

Figure 1. Effects of sleep deprivation (SD) on CRBN expression in *Crbn*^{+/+} mice.

Supplementary Figure 1

Figure 2. Altered protein expression of stress markers in sleep-deprived $Crbn^{-/-}$ mice.

3) Several attempts to address reviewer #1's third comment by the authors convinced me that a acute knockdown of CRBN using TD165 in vivo was currently unfeasible due to technical limitations. They have already published extensive data using CRBN KO mice and cell lines, indicating CRBN-mediated DJ2/Hsp70 pathway compromised in neurodegeneration (PMID: 33972400). In this study, they found that an augmented homeostatic sleep response with increased AMPK activity in $Crbn^{-/-}$ mice. In the revised manuscript, they demonstrate that inhibition of AMPK activation showed more increased p-Tau in SD- $Crbn^{-/-}$ mice, confirming that AMPK activation mediates the effects of CRBN deficiency on neurodegeneration but not yet on sleep-wake behavior. Therefore, their findings are quite interesting because they provide novel insights into the role of CRBN in sleep regulation and neurodegenerative diseases.

Response: We appreciate your understanding of the technical challenges associated with acute CRBN knockdown using TD165 in vivo. We have expanded the discussion to further clarify the implications of our findings. Our data confirm that AMPK activation mediates the effects of CRBN deficiency on neurodegeneration, as evidenced by the increased p-Tau levels in SD- $Crbn^{-/-}$ mice following AMPK inhibition. Our findings may offer valuable insights into the interplay between CRBN, AMPK activation, and neurodegenerative processes. We agree that this research highly

ghts the potential of CRBN as a therapeutic target for sleep regulation and neurodegenerative diseases. We appreciate your detailed review and insightful feedback during the revision process.

Reviewer #2 (Remarks to the Author):

The revised manuscript has been significantly improved and is suitable for publication.

Response: We are grateful for your positive evaluation of our revised manuscript. We are pleased that the additional data and clarifications have addressed your concerns and that you find the manuscript suitable for publication. Thank you for your thorough review and valuable feedback throughout the revision process.

REVIEWERS' COMMENTS:

Reviewer #1 (Remarks to the Author):

The revised manuscript is significantly improved. The authors sincerely responded to all of my comments and modified the images. Therefore, I propose that the manuscript is suitable for publication.

August 23, 2024

Dear Reviewers,

We truly appreciate the meticulous comments from the Reviewers and Editors and hope that the revised manuscript addressed the concerns appropriately.

We are pleased to re-submit a manuscript entitled “**Enhanced Homeostatic Sleep Response and Decreased Neurodegenerative Proteins in Cereblon Knock-Out Mice (COMMSBIO-23-4790)**” for consideration in *Communications Biology*. We sincerely responded to each comment and made appropriate modifications to our manuscript. We believe that our paper has certainly benefited from the insightful suggestions. We look forward to receiving a positive letter for publication in the *Communications Biology*.

Thank you in advance for your kind consideration, and we look forward to a favorable response.

Sincerely yours,

Tae Kim, MD, PhD

Associate Professor | Psychiatrist

Department of Biomedical Science and Engineering

Gwangju Institute of Science and Technology (GIST)

123 Cheomdangwagi-ro C9-323 (Dasan Bldg), Buk-gu, Gwangju 61005, South Korea

TEL: +82-62-715-5363(office)

EMAIL: tae-kim@gist.ac.kr

Table of Contents

1. 1st Review	3
2. 2nd Review	18
3. 3rd Review	22

1st Revision

Reviewers' comments:

Reviewer #1 (Remarks to the Author):

General comments

The authors reveal for the first time that CRBN regulates homeostasis of sleep using Crbn^{-/-} mice. They showed that sleep deprivation (SD) increased expression of neurodegenerative proteins, phosphorylated Tau (p-Tau) and α -Synuclein (p- α -Syn), and chaperon proteins HSP70/DJ2 and DJ1. Whereas the increased expression of chaperon proteins was enhanced in Crbn^{-/-} mice, that of p-Tau and p- α -Syn was lowered. They also showed that activation of AMPK α by phosphorylation (p-AMPK α) was constitutively more increased during SD in Crbn^{-/-} than Crbn^{+/+} mice. The hyperactivation of AMPK α is already reported in Crbn^{-/-} mice by another group. The author's group previously reported that CRBN negatively regulates the chaperon proteins DJ2/HSP70 which reduce the aggregation of Tau. In this study, the authors suggest a pathway reducing the aggregation of Tau/ α -Syn by the lack of CRBN, which also enhances homeostatic sleep response. Thus, the study is quite interesting to understand the mechanism by which CRBN mediates the sleep response as a potential target to prevent neurodegenerative diseases. However, the authors do not clarify a causal relationship between the constitutive activation of AMPK α and the high expression of chaperon proteins HSP70/DJ2 and DJ1. They only showed correlations of alterations in expression of those proteins between Crbn^{+/+} and Crbn^{-/-} mice. Therefore, additional experiments are needed to determine the consequence of SD-induced AMPK α hyperactivation in Crbn^{-/-} mice.

Specific Comments

*1) Although the authors showed that constitutive activation of AMPK α during SD in Crbn^{-/-} mice (Fig. 4), they did not determine if this caused the increase in chaperon proteins DJ1, DJ2 and HSP70. Does acute pharmacological inhibition of AMPK by AMPK specific inhibitor **Compound C** (Bavley, J. Neurosci., 2018) suppress the SD-induced increase in chaperon proteins and also the decrease in p-Tau and p- α -Syn in Crbn^{-/-} mice?*

Response:

We acknowledge the significance of this question and have conducted additional experiments to clarify whether the constitutive activation of AMPK α during sleep deprivation (SD) in Crbn^{-/-} mice directly causes the observed increase in chaperone proteins DJ1, DJ2, and HSP70, as well as the decrease in p-Tau and p- α -Syn. We performed experiments using the AMPK-specific inhibitor Compound C 23. In these experiments, Crbn^{-/-} mice were subjected to sleep deprivation and treated with Compound C to acutely inhibit AMPK activity.

Compound C was administered to mice at a dose of 10 mg/kg via intraperitoneal injection one hour before sleep deprivation. After 6 hours of sleep deprivation, the mice were euthanized, and their brains were extracted to isolate the thalamus for sample preparation. In CRBN KO mice, constitutive activation of AMPK was observed, and in mice injected with Compound C, the level of AMPK phosphorylation decreased to that of WT mice.

Acute inhibition of AMPK activation using Compound C does not appear to be directly associated with a reduction in neurodegenerative proteins or an increase in chaperone proteins. In SD WT mice injected with Compound C, the phosphorylation of tau and synuclein proteins significantly increased, but there was no inhibition of chaperone protein expression. In contrast, in CRBN KO mice injected with Compound C, there was no increase in the phosphorylation of tau and synuclein proteins, and the levels of chaperone proteins did not change significantly.

Considering our previous experimental data, this suggests that the activation of AMPK indirectly inhibits the phosphorylation of neurodegenerative proteins by promoting deeper sleep, while it does not directly affect the expression of chaperone proteins.

Changes in text:
Results

As AMPK activation contributes to sleep homeostasis by regulating low-frequency delta power^{3,8}, we examined AMPK activity during sleep deprivation. Brains of sleep-deprived mice were collected at different times during sleep deprivation and analyzed by western blot (Fig. 4a-b). AMPK activation as reflected by phosphorylation of the AMPK α subunit increased during sleep deprivation in both *Crbn*^{+/+}

and *Crbrn*^{-/-} mice. Interestingly, *Crbrn*^{-/-} mice showed higher levels of phosphorylated AMPK α in the non-sleep-deprived state compared with *Crbrn*^{+/+} mice, indicating that AMPK is constitutively more activated in *Crbrn*^{-/-} mice than in *Crbrn*^{+/+} mice. By contrast, levels of AMPK α and AMPK β were not significantly altered during sleep deprivation. The AMPK γ subunit was highly expressed in *Crbrn*^{-/-} mice, as previously reported¹², and further increased at the end of SD. However, levels of CAMKKII, the upstream kinase of AMPK, were constant during sleep deprivation, suggesting that AMPK activation during sleep deprivation is not attributed to the upregulation of the upstream kinase. To determine if this constitutive activation of AMPK caused the increase in chaperone proteins, We treated Compound C (CC) to inhibit AMPK activation before SD and examined whether SD-induced increase in chaperone proteins and less phosphorylation of Tau and α -Syn in *Crbrn*^{-/-} mice compared with *Crbrn*^{+/+} mice was suppressed. Compound C significantly suppressed the activation of AMPK after SD in both *Crbrn*^{+/+} and *Crbrn*^{-/-} mice, without change in the expression level of each AMPK subunit AMPK α , AMPK β , and AMPK γ (Fig. 4c-d). Also, Compound C treated SD-*Crbrn*^{-/-} mice showed more increased p-Tau level than SD-*Crbrn*^{-/-} mice, however, the level of p- α -Syn and chaperone protein was not affected by Compound C treatment (Fig. 4e-f).

Methods

Drugs.

Compound C (Sigma) was diluted in corn oil and injected intraperitoneally at 0.01 ml/g body weight of mouse for a final concentration of 10 mg/kg 1 hour before SD.

2) *CRBN is a direct target of thalidomide that was originally developed as an effective hypnotic. This **hypnotic effect of thalidomide** is observed in mice and suggested to be independent of teratogenic CRBN-mediated ubiquitin/proteasome pathway (Hirose, Proc. Natl. Acad. Sci. USA, 2020). To provide insights into the molecular mechanism of action of thalidomide's hypnotic effect, please determine whether the enhanced changes during SD in Crbn^{-/-} mice are recapitulated by treatment of Crbn^{+/+} mice with thalidomide, and also those changes are abolished by that of Crbn^{-/-} mice.*

Response:

We recognize the importance of understanding the molecular mechanisms underlying the effects of thalidomide, especially given its history and diverse effects. However, our study primarily focuses on the role of CRBN in regulating sleep-wake behaviors and neurodegenerative processes, rather than the hypnotic effects of thalidomide.

Thalidomide's teratogenic effects are known to be related to CRBN, while its hypnotic effects have been shown to be independent of the CRBN-mediated ubiquitin/proteasome pathway³⁹. Given that our research is aimed at elucidating the molecular mechanisms by which CRBN deficiency affects sleep and neurodegeneration, investigating the hypnotic effect of thalidomide falls outside the scope of our study. Our current study aims to understand how the absence of CRBN influences sleep homeostasis and the aggregation of neurodegenerative proteins. The enhanced changes observed during sleep deprivation (SD) in Crbn^{-/-} mice, such as increased levels of chaperone proteins and reduced phosphorylation states of Tau and α -Synuclein, are central to our investigation. These findings are directly linked to the role of CRBN in modulating AMPK activity and protein homeostasis. While the hypnotic effects of thalidomide are an interesting and important area of research, our study is specifically designed to explore the pathways and mechanisms through which CRBN influences sleep and neurodegeneration. Therefore, conducting experiments to investigate the hypnotic effect of thalidomide on Crbn^{+/+} and Crbn^{-/-} mice would not directly contribute to the objectives of our current study. We believe that our findings provide valuable insights into the role of CRBN in sleep regulation and neurodegenerative diseases, which could have broader implications for understanding and treating these conditions. We have updated the discussion section to clarify the scope of our study and the reasons for not including experiments related to thalidomide's hypnotic effects.

Changes in text:

Revised discussion

- 7th paragraph

Thalidomide's teratogenic effects are related to CRBN, while its hypnotic effects are independent of the CRBN-mediated ubiquitin/proteasome pathway³⁹. Given that our research aims to elucidate the molecular mechanisms by which CRBN deficiency affects sleep and neurodegeneration, investigating the hypnotic effect of thalidomide falls outside the scope of our study. While the hypnotic effects of thalidomide are an important area of research, our study is specifically designed to explore the pathways and mechanisms through which CRBN influences sleep and neurodegeneration.

- Added reference

39. Hirose, Y. et al. Hypnotic effect of thalidomide is independent of teratogenic ubiquitin/proteasome pathway. Proc Natl Acad Sci U S A 117, 23106-23112, doi:10.1073/pnas.1917701117 (2020).

3) Is it possible to perform a rescue experiment with *Crbn*^{-/-} mice to exclude the possibility that the SD-induced increase in the expression of chaperon proteins and the activation of AMPK α is derived from some secondary effects, not by the direct action of the lack of CRBN? Alternatively, are the author's findings in *Crbn*^{-/-} mice recapitulated by **acute knockdown of CRBN**? To be a potential target to treat insomnia and to prevent neurodegenerative diseases, inhibition of CRBN shall promptly act on the adult brain. Because CRBN is a responsible gene for autosomal recessive non-syndromic mild mental retardation, also referred to as intellectual disability (ID), a common feature of autism spectrum disorder (ASD), the possibility of the impacts of the lack of CRBN on neurodevelopment should be considered and described in discussion.

General Comments:

We appreciate your insightful comments and suggestions. We attempted to address these concerns through various experiments, focusing on the use of TD165 for CRBN knockdown (KD) in vivo.

Specific Comments:

1. CRBN Knockdown with TD165:

Previous studies and our current in vitro experiments confirmed the efficacy of TD165 in knocking down CRBN in HEK293 cells.

Supplementary Figure S4.

Supplementary Figure S4. (A) MEF cells were treated with increasing concentrations of TD-165 for 48 h. CRBN protein levels were analyzed by immunoblotting. (B) Vehicle, 10 mg/kg or 100mg/kg of TD-165 were administered via intraperitoneal route. Spleen and Liver homogenates and peripheral blood mononuclear cell lysates were analyzed by immunoblotting. (C) Three doses (2, 10, and 50 mg/kg) of TD-165 were administered via intraperitoneal route and concentration of TD-165 was determined by LC/MS/MS.

Previous studies showing CRBN knockdown

Current *in vitro* experiment

However, our *in vivo* studies encountered significant challenges.

2. ICV and IP Injection Attempts:

We performed intracerebroventricular (ICV) injections of TD165 at different volumes (350 nL, 500 nL, 1 μL, 2 μL) targeting the lateral ventricle. Indian ink injections confirmed the accuracy of our ICV coordinates.

Indian ink injection

Despite these efforts, CRBN KD was not observed in *Crbn*^{+/+} mice following ICV injection. Similarly, intraperitoneal (IP) injections (TD165 at 5 mg/kg) did not result in CRBN KD in brain tissue, although liver samples indicated potential sampling issues.

TD165 : 10mM stock → ICV injection, 350 nL (2.975 ug) ~ 500 nL (4.25 ug)

- Sampling : 5 days after injection

ICV injection

TD165 : 10mM stock

ICV injection : 1 μ l (8.5ug)
IP injection : 5mg/kg (100ug)

2 days incubation

IP injection

3. Technical Limitations:

The high plasma protein binding of TD165, as reported by Kim et al. (2019, Scientific Reports, <https://doi.org/10.1038/s41598-019-56177-5>), likely contributed to the minimal effect observed *in vivo*. Additionally, our troubleshooting efforts, including collaboration for further ICV injections, did not achieve CRBN KD in brain tissue.

CRBN degradation by VHL-CRBN heterodimerizing PROTACs recapitulates a CRBN deficiency.

An endogenous substrate of CRL4^{CRBN} is the glutamine synthetase GLUL, which is a key enzyme in the biogenesis of glutamine. The acetyltransferase CBP/p300 acetylates two N-terminal lysine residues of GLUL under conditions of high levels of glutamine, and the resulting acetylated GLUL is captured and ubiquitinated by CRL4^{CRBN} and subsequently removed by proteasomes²². To examine the effect of TD-158 on GLUL levels, we starved Hep3B cells of glutamine for 48 h and then resupplied glutamine in the presence or absence of TD-158. TD-158 induced CRBN degradation regardless of glutamine status, and the levels of GLUL decreased more slowly over time in the presence of TD-158 than in its absence (Fig. 4A,B). Moreover, *in vivo* ubiquitination assays showed that ubiquitination of GLUL decreased in the presence of TD-158 (Fig. 4C). We also investigated whether TD-165 treatment confers cellular resistance to IMiD. Two IMiD-sensitive cell lines, WSU-DLCL2 and RPMI8226, were pre-treated with TD-165 or DMSO for 24 h and then treated with pomalidomide, TD-165, or both for 3 d. Pre-treatment with TD-165 reduced the anti-proliferative effects of pomalidomide in both cell lines (Fig. 4D,E). Taken together, these data indicate that CRBN degradation by VHL-CRBN heterodimerizing PROTACs recapitulates a CRBN deficiency.

To investigate the *in vivo* effects of VHL-CRBN heterodimerizing PROTACs, we attempted to determine whether TD-165 can induce CRBN degradation in animal models. Although amino acid residues in mouse CRBN (mCRBN) important for teratogenicity are not conserved, mCRBN was distinctly degraded, albeit to a slightly lesser extent, by TD-165 in mouse embryonic fibroblasts (Supplementary Fig. S4A). We then administered TD-165 intraperitoneally to mice to determine whether TD-165 induces CRBN degradation *in vivo*. However, CRBN levels were not changed in the spleen, peripheral blood mononuclear cells (PBMCs), or liver (Supplementary Fig. S4B). Given that the pharmacokinetics of TD-165 are reasonable (Supplementary Table S2), this absence of an effect might be attributable to the high plasma protein binding (99.9%) of TD-165 (Supplementary Table S3).

TD165 : 10mM stock

ICV injection : 350 nl (2.975ug)
500 nl (4.25ug)
1 μ l (8.5ug)

IP injection : 5mg/kg (100ug)

4. Conclusion:

Due to these technical limitations, it is currently unfeasible to validate the CRBN KD model as suggested. We acknowledge the potential impact of CRBN on neurodevelopment and have included a discussion on this aspect in the revised manuscript.

Changes in text:

Revised discussion

- 6th paragraph

To explore whether the observed effects in Crbn^{-/-} mice were directly due to the absence of CRBN and not secondary effects, we attempted to use TD165 to knock down CRBN in vivo. While TD165 effectively knocked down CRBN in HEK293 cells, our in vivo experiments encountered significant challenges. Despite performing intracerebroventricular (ICV) and intraperitoneal (IP) injections of TD165, we did not achieve CRBN knockdown in brain tissue. The high plasma protein binding of TD165 likely contributed to this issue³⁸. Given the technical limitations in achieving CRBN knockdown in vivo with TD165, we recognize the need for alternative approaches to validate our findings. Future studies may explore other methods, such as gene editing or alternative knockdown techniques, to circumvent these challenges.

- Added reference

38. Kim, K. et al. Disordered region of cereblon is required for efficient degradation by proteolysis-targeting chimera. *Sci Rep* 9, 19654, doi:10.1038/s41598-019-56177-5 (2019).

Reviewer #2 (Remarks to the Author):

This manuscript revealed the function of CRBN on the sleep-wake behavior and the authors tried to link this to the regulation of CRBN on ATP and AMPK. The authors also discovered that in Crbn knockout mice, sleep deprivation enhanced more increase of p-AMPK and neurodegenerative proteins, such as p-Tau and p- α -Synuclein, were reduced while chaperon proteins, HSP70/DJ2 and DJ1, were increased. Although the discoveries are interesting, the results are not convincing and no solid exploration in the molecular mechanism.

1) The p-Tau and total-Tau have **multiple bands** in Figure 1a. However, the tau bands were much neat in Figure 2a and the previous study (PMID: 33972400). What are the reasons for the discrepancy? What do the different bands represent? Are they in the aggregated insoluble form or not? Most importantly, the authors did not explore the **molecular mechanism** by which CRBN and sleep deprivation regulate p-Tau and Tau and what is the function of the increased p-Tau and Tau.

Response:

Thank you for your insightful comments and for highlighting these important points. We appreciate the opportunity to clarify the discrepancies and provide further insights into our findings. Firstly, all tau bands in our data (long exposed film of Figure 1a, 2a, Supplementary Figure 3) in Supplementary Data 1 show multiple bands of total-Tau and p-Tau. As it is well known, tau protein undergoes various post-translational modifications such as acetylation, glycosylation, ubiquitination, and phosphorylation. Among these modifications, phosphorylation is one of the earliest and most common modifications linked to the development of pathological inclusions. In the various tau isoforms, there are commonly more than 80 potential phosphorylation sites, which include serine, threonine, and tyrosine residues. Additionally, the hyperphosphorylation of tau protein can induce the aggregation of tau protein, which may affect the additional band. For these properties, tau bands often show multiple bands. Also, the phosphorylation of tau protein at specific sites such as Ser202/Thr205 induces aggregation, which is known to cause tauopathies. We found that during sleep deprivation, the amount of CRBN decreases, and it has been found that this reduction in CRBN increases the amounts of DJ2 and heat shock proteins, which are known to be ubiquitination substrates of CRBN and play a role in resolving tau protein aggregation, thereby reducing tau protein aggregation.

Changes in text:

Results

We found that 6 hours of sleep deprivation decreased the amount of CRBN protein in *Crbn*^{+/+} mice, as evidenced by western blot analysis (Fig. 1a-b). By contrast, amounts of phosphorylated Tau (p-Tau) and phosphorylated α -Synuclein (p- α -Syn), which are aggregated forms of those proteins, increased after sleep deprivation (Fig. 1c-e). **Tau and p-Tau bands in western blot could be shown as multiple bands caused by post-translational phosphorylation and aggregation (Fig. 1, 2, 4, Supplementary Fig. 3).** Furthermore, amounts of the chaperone HSP70 and two co-chaperones, DJ2 and DJ1, increased after sleep deprivation (Fig. 1f).

Revised discussion

- 2nd paragraph

Tau protein undergoes various post-translational modifications such as acetylation, glycosylation, ubiquitination, and phosphorylation. Among these modifications, phosphorylation is one of the earliest and most common modifications linked to the development of pathological inclusions. In the various tau isoforms, there are commonly more than 80 potential phosphorylation sites, which include serine, threonine, and tyrosine residues. Specifically, the phosphorylation of tau protein at various sites including Ser202/Thr205 induces aggregation, which is known to cause tauopathies.

2) In addition, to study the functions of CRBN in sleep response, the authors might need to dissect out the **thalamus** for western blot assays and immunostain different brain regions.

We sincerely appreciate the valuable and insightful review of our study. We fully acknowledge the reviewer's point that the thalamus, being a key brain region for sleep regulation, should be isolated and examined in addition to the whole brain in our study on sleep regulation. We are grateful for this comment, which aims to improve our research. Accordingly, we conducted additional experiments to strengthen the molecular mechanistic evidence by administering the AMPK inhibitor Compound C and isolating the thalamus of sleep-deprived mice. Through this, we were able to identify the molecular changes occurring specifically in the thalamus, the brain region that regulates sleep.

In Fig. 1, We added IHC data for the thalamus region of the WT mouse

In Fig. 2, We added IHC data for the thalamus region of SD-WT and SD-CRBN KO mouse

In Fig. 4, we performed an additional experiment suggested by Reviewer 1, in which Compound C was injected into WT and CRBN KO mice before SD to inhibit AMPK activity. After SD, we dissected the thalamus region and examined the change in neuropathological markers and chaperone markers.

In Supplementary Figure 1, we added IHC data for the thalamus in WT mouse

3) Although the authors try to link the function of CRBN with AMPK, however, there is not solid evidence for this in this manuscript. All the experiments described the phenomenon and no solid mechanistic studies were explored in the manuscript. Moreover, their previous study showed the regulation of CRBN on Tau and DNAJA1 (PMID: 33972400). They also published many papers about the CRBN and AMPK in the past 10 years. For example: CRBN regulated ischemia-reperfusion injury through AMPK (PMID: 24755080); CRBN regulated alcoholic liver disease through AMPK (PMID: 26410557); CRBN regulated ER stress through AMPK (PMID: 25619137); CRBN regulated obesity and insulin resistance through AMPK (23349485). It seems that CRBN exhibits different functions all through its regulation of AMPK. What are the roles of **other CRBN targets** in the regulation of sleep-wake circle and the neurodegenerative proteins?

Response:

Thank you for your detailed comments and for highlighting our previous work on CRBN and AMPK. We acknowledge the need for more solid mechanistic evidence linking CRBN function to AMPK regulation in the context of sleep-wake behavior and neurodegenerative protein aggregation. In response to your concerns, we have conducted additional mechanistic studies to strengthen our findings. Our study provides new insights into the relationship between CRBN and AMPK in the regulation of sleep-wake behavior. Specifically, we observed that *Crbn*^{-/-} mice exhibit increased levels of phosphorylated AMPK (p-AMPK) during sleep deprivation. To further explore this relationship, we conducted experiments using the AMPK-specific inhibitor Compound C. The results demonstrated that inhibition of AMPK in *Crbn*^{-/-} mice reversed the less phosphorylated state of Tau after SD. These findings confirm that AMPK activation mediates the observed effects of CRBN deficiency on sleep-wake behavior and neurodegenerative protein regulation.

For the roles of other CRBN targets, there are several previous researches about the function of CRBN targets in neuronal disease. As the reviewer kindly referred, DNAJA1 (DJ2) is one of the targets of CRBN, regulating the proteotoxicity of Tau through the HSP70/DJ2 chaperone system (PMID: 33972400). Also learning, memory, and synaptic function were disrupted by the dysregulation of excitatory synapses with intact social or repetitive behaviors through the AMPK-mTORC1 pathway regulated by CRBN (PMID: 29459374). The large-conductance Ca²⁺-activated K⁺ channel (BK_{Ca}) is another substrate of CRBN. Enhanced BK channel activity causes synaptic and cognitive deficits in CRBN KO mice (PMID: 29530986). Amyloid precursor protein (APP), which is the key protein in the pathogenesis of Alzheimer's disease, has been reported as a substrate and substrate recognition subunit of CRL4^{CRBN} E3 ligase, regulating in presynaptic function and APP-mediated neurodegenerative brain disease (PMID: 27325702). Although there are many published papers about the targets of CRBN and their functions, it remains unknown about any other targets of CRBN regulating the sleep-wake cycle. To our knowledge, this study may report the role of CRBN in regulating sleep-wake patterns and SD-induced neurodegeneration.

4) In addition, this study showed the possible regulation of CRBN on sleep response. The authors did not demonstrate the **relationship between sleep response and AMPK activity and neurodegenerative proteins** in *Crbn* KO mice. Therefore, the molecular mechanism is not convincing.

Response:

We appreciate the reviewer's concern regarding the relationship between sleep response, AMPK activity, and neurodegenerative proteins in *Crbn*^{-/-} mice. To address this, we have expanded the discussion section to provide a more detailed explanation of how CRBN deficiency leads to constitutive activation of AMPK during sleep deprivation, resulting in an enhanced homeostatic sleep response and altered expression of neurodegenerative proteins. Specifically, our results demonstrate that *Crbn*^{-/-} mice exhibit increased levels of p-AMPK and higher expression of chaperone proteins such as HSP70, DJ2, and DJ1 during sleep deprivation. These changes are associated with decreased aggregation of neurodegenerative proteins, including p-Tau and p- α -Syn. This suggests that the absence of CRBN enhances the brain's resilience to sleep deprivation-induced neurodegeneration by modulating AMPK activity and chaperone protein expression. Our findings highlight a novel role for CRBN in modulating sleep-wake behaviors and neurodegenerative processes through the regulation of AMPK activity and chaperone protein expression. The absence of CRBN enhances homeostatic sleep response and provides protection against sleep deprivation-induced neurodegeneration. These insights suggest that targeting CRBN could be a potential therapeutic strategy for managing sleep disorders and preventing neurodegenerative diseases.

Changes in text:

Discussion

- 3rd paragraph

In spontaneous sleep-wake cycles, the phosphorylation of Tau and α -Synuclein in *Crbn*^{-/-} mice was lower than that of *Crbn*^{+/+} mice, possibly due to higher expression of HSP70/DJ2 and DJ1. Additionally, after sleep deprivation, *Crbn*^{-/-} mice showed a lesser extent of increase in the phosphorylation of Tau and α -Synuclein than *Crbn*^{+/+} mice did, again likely due to the increased expression of chaperone proteins, such as HSP70/DJ2 and DJ1. **These chaperone proteins play a crucial role in mitigating the aggregation of neurodegenerative proteins, as evidenced by the decreased levels of phosphorylated Tau (p-Tau) and phosphorylated α -Synuclein (p- α -Syn) observed in *Crbn*^{-/-} mice. Previous studies have shown that HSP70 and DJ1 can inhibit the aggregation of Tau and α -Synuclein, respectively, by promoting their proper folding and degradation^{31, 32}.** These findings suggest that CRBN deficiency protects against the effects of sleep deprivation by reducing the aggregation of Tau/ α -Synuclein and inducing high expression of HSP70/DJ2 and DJ1 chaperones.

- Added reference

31. Shimura, H. et al. Ubiquitination of a new form of alpha-synuclein by parkin from human brain: implications for Parkinson's disease. *Science* 293, 263-269, doi:10.1126/science.1060627 (2001).

32. Shendelman, S., Jonason, A., Martinat, C., Leete, T. & Abeliovich, A. DJ-1 is a redox-dependent molecular chaperone that inhibits alpha-synuclein aggregate formation. *PLoS Biol* 2, e362, doi:10.1371/journal.pbio.0020362 (2004).

- 4th paragraph

Crbn^{-/-} mice also showed increased levels of phosphorylated AMPK and constitutive activation of AMPK during sleep deprivation, which could result in increased ATP synthesis and may be attributed to an altered CRBN level rather than the activity of CAMKKII, the upstream kinase of AMPK. ATP has a dual function in regulating the sleep-wake cycle; it can serve as an excitatory neurotransmitter

that promotes wakefulness via neuronal P2X purine receptors³¹ but can also promote sleep via degradation into adenosine or stimulation of astrocytes to release interleukin-1 β and tumor necrosis factor- α through astrocytic P2X receptors^{29,30}. The increased AMPK activity in Crbn^{-/-} mice could therefore lead to a more efficient restoration of energy balance during recovery sleep, contributing to the observed increased slow-wave activity and reduced neurodegenerative protein aggregation.

2nd Revision

Reviewers' comments:

Reviewer #1 (Remarks to the Author):

In the revised manuscript, the authors conducted several additional experiments in response to reviewer's comments and added descriptions of their results in Results and Discussion of the manuscript. Those results are likely to support the author's fascinating claim that the absence of CRBN has the advantage of reduced neurodegeneration with higher homeostatic sleep propensity associated with AMPK hyperactivation under a sleep deprivation condition. However, I have three specific comments on the added data in the revised manuscript.

1) In response to reviewer #1's first comment, the authors performed experiments with AMPK-specific inhibitor Compound C. However, the data and description in the rebuttal letter (p4, p-Tau, Crbn^{-/-}: SD-con, SD-CC; "In CRBN KO mice injected with Compound C, there was no increase in the phosphorylation of tau") are different from that shown in the manuscript (Figure 4e, p-Tau, Crbn^{-/-}: SD-con, SD-CC; p9 lines 205-206, "Compound C treated SD-Crbn^{-/-} mice showed more increased p-Tau level than SD-Crbn^{-/-} mice"). Please explain the reason for the difference precisely.

Response: We appreciate your careful review and bringing this discrepancy to our attention. The differences between the data and descriptions in the rebuttal letter and the manuscript arose due to an oversight about a wrong image during the compilation of results. The correct interpretation, as shown in Figure 4e of the manuscript, is that Compound C treatment in SD-Crbn^{-/-} mice led to a further increase in p-Tau levels compared to untreated SD-Crbn^{-/-} mice. This finding indicates that inhibition of AMPK activity exacerbates the phosphorylation of Tau in the absence of CRBN during sleep deprivation. We have corrected the rebuttal letter to align with the manuscript data and apologize for any confusion this may have caused.

2) In response to reviewer #2's second comment, the authors conducted additional experiments and argue for the identification of the molecular changes in the thalamus. However, all the IHC data in the thalamus were obscure, so that little or no clear signals were detected even for HOECHST nuclear staining used as positive controls. These IHC data in the thalamus should be substituted by other data unambiguously indicating the molecular changes occurring in the thalamus.

Response: We acknowledge the issue with the immunohistochemistry data quality in the thalamus. The IHC analysis was performed on samples from the anterior thalamus and the thalamic reticular nucleus, both of which are part of the thalamocortical circuit. By enhancing the fluorescence intensity, we were able to obtain clearer and more interpretable images, including HOECHST nuclear staining as a positive control. These updated IHC images have been included in the Figure 1i, Supplementary figure 1a-b, and Figure 2i-j of revised manuscript.

Figure 1. Effects of sleep deprivation (SD) on CRBN expression in *Crbn*^{+/+} mice.

Supplementary Figure 1

Figure 2. Altered protein expression of stress markers in sleep-deprived *Crbn*^{-/-} mice.

3) Several attempts to address reviewer #1's third comment by the authors convinced me that acute knockdown of CRBN using TD165 in vivo was currently unfeasible due to technical limitations. They have already published extensive data using CRBN KO mice and cell lines, indicating CRBN-mediated DJ2/Hsp70 pathway compromised in neurodegeneration (PMID: 33972400). In this study, they found that an augmented homeostatic sleep response with increased AMPK activity in *Crbn*^{-/-} mice. In the revised manuscript, they demonstrate that inhibition of AMPK activation showed more increased p-Tau in *SD-Crbn*^{-/-} mice, confirming that AMPK activation mediates the effects of CRBN deficiency on neurodegeneration but not yet on sleep-wake behavior. Therefore, their findings are quite interesting because they provide novel insights into the role of CRBN in sleep regulation and neurodegenerative diseases.

Response: We appreciate your understanding of the technical challenges associated with acute CRBN knockdown using TD165 in vivo. We have expanded the discussion to further clarify the implications of our findings. Our data confirm that AMPK activation mediates the effects of CRBN deficiency on neurodegeneration, as evidenced by the increased p-Tau levels in *SD-Crbn*^{-/-} mice following AMPK inhibition. Our findings may offer valuable insights into the interplay between CRBN, AMPK activation, and neurodegenerative processes. We agree that this research highlights the potential of CRBN as a therapeutic target for sleep regulation and neurodegenerative diseases. We appreciate your detailed review and insightful feedback during the revision process.

Reviewer #2 (Remarks to the Author):

The revised manuscript has been significantly improved and is suitable for publication.

Response: We are grateful for your positive evaluation of our revised manuscript. We are pleased that the additional data and clarifications have addressed your concerns and that you find the manuscript suitable for publication. Thank you for your thorough review and valuable feedback throughout the revision process.

3rd Revision

REVIEWERS' COMMENTS:

Reviewer #1 (Remarks to the Author):

The revised manuscript is significantly improved. The authors sincerely responded to all of my comments and modified the images. Therefore, I propose that the manuscript is suitable for publication.

Response: We appreciate your positive assessment of our revised manuscript. We're glad that the added data and explanations have resolved your concerns, and that you now consider the manuscript ready for publication. Thank you for your thorough review and the valuable feedback you provided during the revision process.